# Stability in Training PINNs for Stiff PDEs: Why Initial Conditions Matter

## Abstract

Training Physics-Informed Neural Networks (PINNs) on stiff time-dependent PDEs remains highly unstable. Through rigorous ablation studies, we identify a surprisingly critical factor: the enforcement of initial conditions. We present the first systematic ablation of two core strategies, hard initial-condition constraints and adaptive loss weighting. Across challenging benchmarks (sharp transitions, higher-order derivatives, coupled systems, and high frequency modes), we find that exact enforcement of initial conditions (ICs) is not optional but essential. Our study demonstrates that stability and efficiency in PINN training fundamentally depend on ICs, paving the way toward more reliable PINN solvers in stiff regimes.

## 1 Introduction

Time-dependent partial differential equations (PDEs) constitute the mathematical foundation of numerous scientific and engineering disciplines, serving as essential tools for modeling a diverse range of physical phenomena. Their applications span from the intricate dynamics of fluid flow, as described by the Navier–Stokes equations, to the complex interplay of chemical reactions in biological systems, captured by reaction–diffusion models (Whitham, 2011; Burgers, 1948; Allen & Cahn, 1979; Shen & Yang, 2010; Bazant, 2017; Horstmann et al., 2013; Tian et al., 2015; Hyman et al., 2014; Takatori & Brady, 2015; Cahn & Hilliard, 1958; Lee et al., 2014; Miranville, 2017; Kudryashov, 1990; Michelson, 1986; De Kepper et al., 1991; Nishiura & Ueyama, 1999; Zhabotinsky, 2007; Cassani et al., 2021; Kato, 1987; Kevrekidis et al., 2001; Fibich, 2015). However, solution of stiff time-dependent PDEs can be exceedingly challenging to obtain and often requires substantial computational resources and specialized numerical techniques.

Machine learning methodologies have demonstrated remarkable success across various scientific and engineering fields, transforming areas such as protein folding prediction (Jumper et al., 2021), drug discovery and development (Carracedo-Reboredo et al., 2021), and even the optimization of fundamental algorithmic processes such as matrix-vector multiplication (Fawzi et al., 2022). However, traditional machine learning approaches, which rely on large datasets, can be impractical in scientific contexts where data acquisition is expensive, time consuming, or simply infeasible. Moreover, purely data-driven models often lack physical fidelity and interpretability, as they may fail to capture the underlying physical principles governing the systems they aim to describe. Physics-informed machine learning (PIML), and in particular physics-informed neural networks (PINNs), address these limitations by integrating physical laws directly into the learning process. This significantly reduces the amount of training data required. By embedding constraints derived from governing PDEs into the neural network architecture, PINNs enable accurate predictions even when data are sparse or noisy (Yang et al., 2019; Chen et al., 2021; Raissi & Karniadakis, 2018; Raissi et al., 2019; Liu & Wang, 2019; 2021; Jin et al., 2021; Cai et al., 2021; Rad et al., 2020; Zhu et al., 2021; Chen et al., 2022; Wang et al., 2023). Despite their promise, effectively training PINNs for stiff time-dependent PDEs remains a major challenge. The difficulty of propagating information from initial and boundary conditions into the computational domain, combined with the need to balance multiple loss terms, often results in convergence issues and suboptimal solutions (Wang et al., 2020a;b; McClenny & Braga-Neto, 2020; Coutinho et al., 2023).

This paper presents a complete study of investigating the importance of the training components for PINNs on solving stiff time-dependent PDES. First, we conduct an ablation study of training algorithms for PINNs applied to stiff time-dependent PDEs, focusing specifically on two schemes: hard

constraints and self-adaptive loss weights (McClenny & Braga-Neto, 2020). The hard-constraint scheme directly embeds information about initial and boundary conditions into the neural network architecture. This transformation reduces the effects of spectral bias from training a multi-objective loss, leading to improved solution accuracy (Wang et al., 2020b). In addition, we investigate the use of adaptive loss weights to alleviate spectral bias in PINN training (McClenny & Braga-Neto, 2020). Through our ablation study of these training schemes, we conclude that the hard-constraint transformation provides the best improvement. Next, we provide a Neural Tangent Kernel perspective on how the hard-constrained transformation is reducing the spectral bias in PINN training. We emphasize that hard constraints is not the only effective strategies for handling stiff PDEs. Such method can be integrated with other advanced PINN techniques, such as causal PINNs (Wang et al., 2020b), time-marching PINNs (Wight & Zhao, 2020), residual-based-attention (RBA) PINNs (Anagnostopoulos et al., 2024), curriculum training (Krishnapriyan et al., 2021), and co-training PINNs (Zhong et al., 2024), to achieve synergistic improvements in prediction accuracy and robustness.

The rest of the paper is structured as follows. We compare our methodology to other related works in section 1.1; in section 2, we discuss the hard-constrained framework and other training methodologies used in our ablation and comparison study; we presented a detailed discussion on the theoretical mechanism behind the hard-constrained transformation in section 3.1; we presented the results (in summary) from four major categories of examples in section 4; we conclude our paper in 5, pointing out several future directions for this line of research. Since the examples we conducted are too extensive, the additional details are presented in Appendix section D.

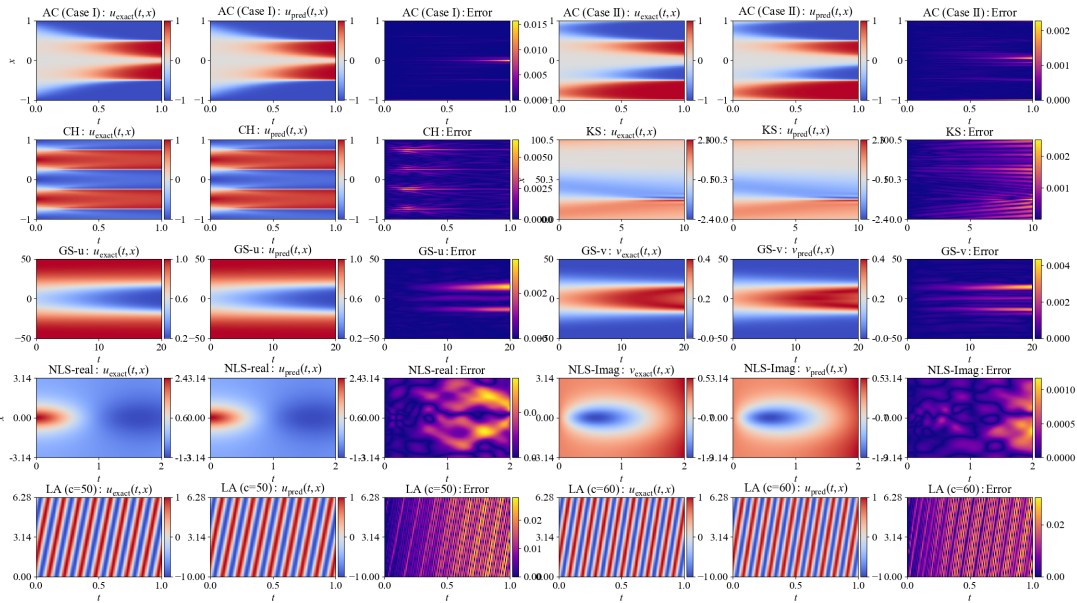

Figure 1: 7 Benchmark Results: Truth vs HC-PINN vs Absolute Error.

## 1.1 RELATED WORK

PINNs have shown significant potential to solve a wide range of PDEs. However, challenges arise, particularly when dealing with "stiff" PDEs, where explicit numerical methods require extremely small time steps for stability. A primary concern is the imbalance between the fitting of initial data and residual components within the PINN loss function (Wight & Zhao, 2020; Shin et al., 2020; Wang et al., 2020a;b). This imbalance often leads gradient descent to prioritize minimizing the residual loss over data fitting loss, hindering convergence to accurate solutions. This issue is exacerbated in forward problems, where data is primarily concentrated at initial and boundary conditions, leaving limited information within the PDE domain. Specifically, for time-dependent

problems, PINNs struggle to propagate information from initial conditions to subsequent time steps, a phenomenon widely documented in the literature (Wight & Zhao, 2020; McClenny & Braga-Neto, 2020; Krishnapriyan et al., 2021; Wang et al., 2022; Haitsiukevich & Ilin, 2022).

Several strategies have been proposed to address these challenges. One approach, introduced by Wight & Zhao (2020), involves time-marching, where the time domain is divided into smaller intervals, with PINNs being sequentially trained on each interval, starting from the initial condition. While effective, this method can be computationally expensive due to the need to train multiple networks. To improve information propagation, researchers have emphasized the importance of weighting initial condition data and residual points near the initial time (Wight & Zhao, 2020; Mc-Clenny & Braga-Neto, 2020). McClenny & Braga-Neto (2020) demonstrated that their self-adaptive PINNs can autonomously learn these weights during training, particularly in scenarios with complex initial conditions, such as the wave equation. Further advancements include the "sequence-to-sequence" approach (Krishnapriyan et al., 2021), causality PINN (Wang et al., 2022), artificial viscosity PINN (Coutinho et al., 2023), Co-training PINN (Zhong et al., 2024).

Beyond loss weighting and time marching, structural modifications to PINNs have also been explored. Braga-Neto (2022) introduced Characteristic-Informed Neural Networks (CINNs) for transport equations, where PDE information is integrated into the neural network architecture. A similar approach, albeit for Dirichlet boundary conditions, was proposed in (Lagaris et al., 1998). Our method builds upon these architectural modifications, with a focus on enforcing (near-)exact fit of the initial condition using hard-constrained transformation.

## 2 METHODOLOGY

We consider the following setup for general time dependent PDEs. Let $u$ be an unknown function defined on $[0, T] \times \bar{\Omega}$ with $\Omega \subset \mathbb{R}^d$. Here the physical domain $\Omega$ comes with a Lipschitz boundary $\partial\Omega$ and $\bar{\Omega} = \Omega \cup \partial\Omega$. And we denote $\mathbf{x} = \begin{pmatrix} x_1 & \cdots & x_d \end{pmatrix}^\top \in \Omega$. Then we say that $u$ is a solution of a time-dependent PDE, if $u$ satisfies the following

$$\begin{cases} u_t(t, \mathbf{x}) + \mathcal{P}[u](t, \mathbf{x}) &= f(t, \mathbf{x}), \quad (t, \mathbf{x}) \in (0, T] \times \Omega, \\ u(0, \mathbf{x}) &= u_0(\mathbf{x}), \quad \mathbf{x} \in \bar{\Omega}, \\ \mathcal{B}[u](t, \mathbf{x}) &= g(t, \mathbf{x}), \quad (t, \mathbf{x}) \in [0, T] \times \partial\Omega, \end{cases} \tag{1}$$

Here the operator $\mathcal{P}$ has order $K$ and it is defined as

$$\mathcal{P}[u] = h(u, \{\partial^k_{x_1^{j_1} x_2^{j_2} \cdots x_d^{j_d}} u\}_{k=1}^K), \quad h \text{ is any multivariate function,}$$

where $\partial^k_{x_1^{j_1} x_2^{j_2} \cdots x_d^{j_d}} u$ is a set that contains all of partial derivatives of $u$ with respective to $\mathbf{x}$ of $k^{th}$ order in the multi variate sense, i.e., for the powers of each coordinate, we have

$$j_1, j_2, \cdots, j_d \geq 0 \quad \text{and} \quad j_1 + j_2 + \cdots + j_d = k.$$

Furthermore $\mathcal{B}$ is an operator defined for $(t, \mathbf{x})$ on the boundary. We also assume that the functions $u_0$, $f$, and $g$ are user inputs that satisfy the desired regularity so that the existence and uniqueness of solutions for such a PDE are guaranteed. We also assume that the compatibility condition, $g(0, \mathbf{x}) = u_0(\mathbf{x})$ for $\mathbf{x} \in \partial\Omega$, is satisfied to prevent any ill-conditioning of the solution.

**Types of Boundary Conditions**: one can consider many kinds of boundary conditions as follows

$$\mathcal{B}[u](t, \mathbf{x}; a, b) = au(t, \mathbf{x}) + b\frac{\partial u}{\partial \mathbf{n}}(t, \mathbf{x}), \quad \text{Robin type,}$$

where $\mathbf{n}$ is the outward normal vector to $\partial\Omega$ and $a, b$ are two fixed constants. Special values of $a$ and $b$ can give two other BC cases, such as when $b = 0$ gives $\mathcal{B}[u](t, \mathbf{x}) = u(t, \mathbf{x})$ (Dirichlet) and $a = 0$ gives the $\mathcal{B}[u](t, \mathbf{x}) = \frac{\partial u}{\partial \mathbf{n}}(t, \mathbf{x})$ (Neumann). The periodic boundary condition gives $d$ different equations, and they are,

$$\mathcal{B}[u](t, \mathbf{x}) = u(t, \mathbf{x}) - u(t, \mathbf{x} + P\mathbf{e}_i) = 0, \quad i = 1, \cdots, d. \tag{2}$$

Here $P$ is the period and $\mathbf{e}_i$ is the $i^{th}$ standard basis vector in $\mathbb{R}^d$, that is,

$$\mathbf{e}_i = \underbrace{[0 \quad \cdots \quad 0 \quad 1 \quad 0 \quad \cdots \quad 0]^\top}_{1 \text{ is the in the } i^{th} \text{ position.}}.$$

One can even consider a mixture of two or all of the above BC conditions. For example, we can have $\Omega = \Gamma_1 \cup \cdots \cup \Gamma_K$ with each $\Gamma_i$ defines a different kind of BC. In this paper, we will mainly focus on the periodic type of boundary conditions.

Recent advancements in scientific machine learning integrate the principles of Physics, particularly the Physics-derived PDEs, into the training process of machine learning models. This integration enables the development of PINNs to solve for $u$ as follows: find an approximate solution from a set of deep neural networks $\mathcal{H}_{\text{NN}}$, where each network has the same depth, the same number of neurons on each hidden layer, and the same activation functions on each layer, that minimizes the following loss functional

$$\text{Loss}(u_{nn}) = \text{Data Loss}(u_{nn}) + \lambda * \underbrace{\text{IC Loss}(u_{nn}) + \text{BC Loss}(u_{nn}) + \text{PDE Loss}(u_{nn})}_{\text{Physical Loss}} \quad (3)$$

where $\lambda$ is a regularization parameter, and the two losses are as follows

$$\begin{cases} \text{Data Loss}(u_{nn}) = \frac{1}{N_{Data}} \sum_{i=1}^{N_{Data}} |u_{nn}(t^{Data}, \mathbf{x}_i^{Data}) - u(t^{Data}, \mathbf{x}_i^{Data})|^2 \\ \text{IC Loss}(u_{nn}) = \frac{1}{N_{IC}} \sum_{i=1}^{N_{IC}} |u_{nn}(0, \mathbf{x}_i^{IC}) - u_0(\mathbf{x}_i^{IC})|^2, \\ \text{BC Loss}_{\text{per}}(u_{nn}) = \sum_{i=1}^{d} \frac{1}{N_{BC}} \sum_{j=1}^{N_{BC}} |u_{nn}(t_j^{BC}, \mathbf{x}_j^{BC}) - u_{nn}(t_j^{BC}, \mathbf{x}_j^{BC} + P\mathbf{e}_i)|^2, \\ \quad \text{or} \\ \text{BC Loss}_{\text{gen}}(u_{nn}) = \frac{1}{N_{BC}} \sum_{j=1}^{N_{BC}} |(\mathcal{B}[u_{nn}] - g)(t_j^{BC}, \mathbf{x}_j^{BC})|^2, \\ \text{PDE Loss}(u_{nn}) = \frac{1}{N_{CL}} \sum_{i=1}^{N_{CL}} |(\frac{\partial u}{\partial t} + \mathcal{P}[u_{nn}] - f)(t_i^{CL}, \mathbf{x}_i^{CL})|^2 \end{cases}$$

for $u_{nn} \in \mathcal{H}_{\text{NN}}$. Note that BC Loss$_{\text{per}}(u_{nn})$ enforces continuity and periodicity along each spatial direction $i$, while in BC Loss$_{\text{gen}}(u_{nn})$, $\mathcal{B}$ and $g$ denote the boundary operator (e.g., identity or normal derivative) and the prescribed boundary condition, respectively. Here $\{(t^{CL}, \mathbf{x}^{CL})_i\}_{i=1}^{N_{CL}} \in (0, T] \times \Omega$ are called collocation points, $\{(0, \mathbf{x}^{IC})_i\}_{i=1}^{N_{IC}} \in \{0\} \times \bar{\Omega}$ are the initial condition points, $\{(t^{BC}, \mathbf{x}^{BC})_i\}_{i=1}^{N_{BC}} \in [0, T] \times \partial\Omega$ are the boundary condition points, and $\{(t^{Data}, \mathbf{x}^{Data})_i\}_{i=1}^{N_{Data}} \in (0, T] \times \Omega$ are known data points (they can be noisy). The minimizer, denoted as $u_{NN}$, will an be approximate solution to (1).

## 2.1 HARD CONSTRAINTS

The main motivation for us to consider hard constrains transformation for PINNs is the possible ill-conditioning of the PDE solution operator's dependence on IC and BC. Traditional numerical methods do not encounter this kind of difficulty, as the IC data is exactly satisfied at the starting point of the solution loop. PINNs, on the other hand, use $L^2$-loss to fit the IC/BC and PDE residual data, which causes the solutions to be highly sensitive to the tightness of fit for the IC in the training, as pointed out in (McClenny & Braga-Neto, 2020). By transforming the need of fitting IC/BC conditions out of PINNs, it reduces the spectral bias in the training. Therefore, We consider the following transformation

$$\tilde{u}(t, \mathbf{x}) = \psi(t, \mathbf{x}) + \phi(t, \mathbf{x})u_{nn}(t, \mathbf{x}), \quad (4)$$

Here the functions $\psi$ and $\phi$ are smooth functions with the following property

$$\begin{cases} \psi(0, \mathbf{x}) = u_0(\mathbf{x}), & \phi(0, \mathbf{x}) = 0, & \text{for } \mathbf{x} \in \bar{\Omega}, \\ \psi(t, \mathbf{x}) = g(t, \mathbf{x}), & \phi(t, \mathbf{x}) = 0, & \text{for } (t, \mathbf{x}) \in [0, T] \times \partial\Omega. \end{cases} \quad (5)$$

In the case of periodic BC, we simply require $\psi$ and $\phi$ to be both $P$-periodic. When such transformation is used, then the training for $u^{nn}$ is updated as

$$\text{Loss}_{HC}(u_{nn}) = \frac{1}{N_{CL}} \sum_{i=1}^{N_{CL}} |(\tilde{u}_t + \mathcal{P}[\tilde{u}] - f)(t_i^{CL}, \mathbf{x}_i^{CL})|^2, \quad \tilde{u} = \psi + \phi u_{nn}. \quad (6)$$

We have successfully reduced the multi-objective loss to a single-objective; however the information about IC and BC has been moved into the PDE residual loss through $\psi$ and $\phi$.

**Periodic Neural Networks**: even though we require $\psi$ and $\phi$ to be periodic, we still need to build the information of periodicity into $u_{nn}$, hence we consider the following composition

$$u_{nn}(t, x) = f_{nn}(v(t, x)), \quad \text{where } f_{nn} \text{ is a neural network and } x \in \mathbb{R}.$$

Here $v(t, x) = [t \quad \mathcal{F}_m(x)]^\top$ with

$$\mathcal{F}_m(x) = \begin{bmatrix} 1 & \cos\left(\frac{2\pi}{P}x\right) & \sin\left(\frac{2\pi}{P}x\right) & \cdots & \cos\left(\frac{2\pi}{P}mx\right) & \sin\left(\frac{2\pi}{P}mx\right) \end{bmatrix}$$

where $m$ is a positive integer hyperparameter controlling the number of Fourier modes. With this construction, each trigonometric feature is exactly $P$–periodic, and therefore any linear combination (and hence $u_{nn}$) will inherit $P$–periodicity (Dong & Ni, 2021). For $\mathbf{x} = [x \quad y]$, we can use the following

$$v(t, x, y) = [t \quad \mathcal{F}_m(x) \quad \mathcal{F}_m(y)];$$

similarly for higher dimensional $\mathbf{x} = [x_1 \quad \cdots \quad x_d]$, we have

$$v(t, \mathbf{x}) = [t \quad \mathcal{F}_m(x_1) \quad \cdots \quad \mathcal{F}_m(x_d)].$$

## 2.2 Other Training Enhancements

We also consider mini-batching (to handle large sample size) and self-adaptive loss weights (McClenny & Braga-Neto, 2020), please see sections C.1 and C.2 for details.

## 3 Theoretical Foundation for Hard Constraints

This section provides the theoretical justification for the stability and conditioning advantages of the hard-constraint transformation in PINNs. We divide the analysis into three parts: (1) well-posedness of the hard-constraint transformation, (2) Hessian and NTK characterization, and (3) implications for training decay and spectral bias.

### 3.1 Well-Posedness of the Hard-Constraint Transformation

We start with the minimal regularity conditions on the pair $(\psi, \phi)$ for a well-defined transformation.

**Theorem 1** (Well-Posedness of the Transformation). *If $\phi, \psi \in C^1([0, T]; C^k(\bar{\Omega}))$ and $\phi_t(0, \mathbf{x}) \neq 0$, then the transformed network $u_{nn}$ satisfies the PDE*

$$\begin{cases} \phi\partial_t u_{nn} + \mathcal{P}[\psi + \phi u_{nn}] + \phi_t u_{nn} = f - \psi_t, & (t, \mathbf{x}) \in (0, T] \times \Omega, \\ u_{nn}(0, \mathbf{x}) = \frac{u_t(0, \mathbf{x}) - \psi_t(0, \mathbf{x})}{\phi_t(0, \mathbf{x})}, & \mathbf{x} \in \bar{\Omega}, \end{cases}$$

*and the system is well-posed in the sense that:*

- *$u_{nn}$ is uniquely determined and continuously differentiable in time;*
- *the transformation is non-singular near $t = 0$;*
- *$u(t, \mathbf{x}) = \psi + \phi u_{nn}$ automatically satisfies the initial condition $u(0, \mathbf{x}) = u_0(\mathbf{x})$.*

**Remark 1.** *This theorem ensures existence and uniqueness of the transformed PDE system and guards against degeneracy at the initial time. Intuitively, $\phi$ scales the PDE dynamics and $\psi$ encodes the initial condition, ensuring that the neural approximation evolves from the correct starting manifold. Hence our choice of setting $\psi(t, \mathbf{x}) = e^{-Ct}u_0(\mathbf{x})$ and $\phi(t, \mathbf{x}) = 1 - e^{-Ct}$ ensures boundedness as $T \to \infty$ and stable temporal scaling. Using $\phi = t$ would cause deterioration at large $T$ since the scaling diverges. We will obtain (when $T \to \infty$)*

$$u_{nn}(\infty, \mathbf{x}) = \mathcal{P}^{-1}[f](\infty, \mathbf{x}) = u(\infty, \mathbf{x}).$$

*In the case of $\mathcal{P}$ being linear and $\phi(t, \mathbf{x}) = \phi(t)$, we end up with a much simpler PDE*

$$\phi\partial_t u_{nn} + \phi\mathcal{P}[u_{nn}] + \phi' u_{nn} = f - \psi_t - \mathcal{P}(\psi),$$

*If $\phi(t) \neq 0$ when $t \neq 0$, then it can be simplified further to*

$$\partial_t u_{nn} + \mathcal{P}[u_{nn}] + \frac{\phi'}{\phi}u_{nn} = \frac{f - \psi_t - \mathcal{P}(\psi)}{\phi}$$

*We can see that $u_{nn}$ satisfies a similar PDE to the original one with the additional reaction term $\frac{\phi'}{\phi}u_{nn}$ and updated forcing $\frac{f-\psi_t-\mathcal{P}(\psi)}{\phi}$, which contains the information of $u_0$ and g.*

### 3.2 CONDITIONING AND HESSIAN STRUCTURE

Next, we examine how the transformation affects the loss Hessian that determines training stability.

**Network Representation.** Assume that $\mathcal{P}$ is linear and $\phi(t, \mathbf{x}) = \phi(t) \neq 0$ for $t > 0$, then for a one-hidden layer network,

$$u(\mathbf{y}; \boldsymbol{\theta}) = \mathbf{h}^\top(\mathbf{y})\boldsymbol{\theta}, \quad \mathbf{h} = [h_1(\mathbf{y}) \quad \cdots \quad \mathbf{h}_D(\mathbf{y})]^\top,$$

where $\mathbf{y} = \mathbf{z} = (t, \mathbf{x})$ or $\mathbf{y} = \boldsymbol{\zeta} = (0, \mathbf{x}^{IC})$ are the learnable weights. Here $\mathbf{h}(\mathbf{y})$ denotes the vector of hidden-layer activations (features).

**Loss Formulation.** Therefore the two losses, consistent with the definitions in (3) and (6), can be written in compact matrix form as

$$\text{Loss}_s(u(\mathbf{y}; \boldsymbol{\theta})) = \frac{1}{2N}|\mathbf{A}\boldsymbol{\theta} - \mathbf{f}|^2_{\ell^2(\mathbb{R}^N)} + \frac{1}{2M}|\mathbf{C}\boldsymbol{\theta} - \mathbf{u}_0|^2_{\ell^2(\mathbb{R}^M)}, \quad \text{(soft constrained)}$$

$$\text{Loss}_h(u(\mathbf{z}; \boldsymbol{\theta})) = \frac{1}{2N}|\boldsymbol{\Lambda}_\phi \mathbf{A}\boldsymbol{\theta} + \boldsymbol{\Lambda}_{\phi_t}\mathbf{B}\boldsymbol{\theta} - \tilde{\mathbf{f}}|^2_{\ell^2(\mathbb{R}^N)}, \quad \text{(hard constrained)}$$

where $\mathbf{A}, \mathbf{B}, \mathbf{C}$ collect the PDE, feature, and IC residual operators respectively:

$$\mathbf{A} = \begin{bmatrix} \mathcal{L}_s[\mathbf{h}^\top(\mathbf{z}_1)] \\ \vdots \\ \mathcal{L}_s[\mathbf{h}^\top(\mathbf{z}_N)] \end{bmatrix}, \quad \mathbf{B} = \begin{bmatrix} \mathbf{h}^\top(\mathbf{z}_1) \\ \vdots \\ \mathbf{h}^\top(\mathbf{z}_N) \end{bmatrix}, \quad \text{and} \quad \mathbf{C} = \begin{bmatrix} \mathbf{h}^\top(\boldsymbol{\zeta}_1) \\ \vdots \\ \mathbf{h}^\top(\boldsymbol{\zeta}_M) \end{bmatrix},$$

where $\mathbf{x}_i^{IC} \in \bar{\Omega}$; and for the diagonal matrices $\boldsymbol{\Lambda}_\phi$ and $\boldsymbol{\Lambda}_{\phi_t}$

$$\boldsymbol{\Lambda}_\phi = \text{diag}(\phi(\mathbf{z}_1), \cdots, \phi(\mathbf{z}_N)) \quad \text{and} \quad \boldsymbol{\Lambda}_{\phi_t} = \text{diag}(\phi_t(\mathbf{z}_1), \cdots, \phi_t(\mathbf{z}_N)),$$

and the vectors $\mathbf{f}, \tilde{\mathbf{f}}$, and $\mathbf{u}_0$ contain sampled values of $f$, $f - \mathcal{L}_s[\psi]$, and $u_0$.

**Hessian Expressions.** The corresponding Hessians of these objectives are

$$H_s(\boldsymbol{\theta}) = \frac{1}{N}\mathbf{A}^\top\mathbf{A} + \frac{1}{M}\mathbf{C}^\top\mathbf{C}, \quad H_h(\boldsymbol{\theta}) = \frac{1}{N}(\mathbf{A}^\top\boldsymbol{\Lambda}_\phi^2\mathbf{A} + 2\mathbf{A}^\top\boldsymbol{\Lambda}_\phi\boldsymbol{\Lambda}_{\phi_t}\mathbf{B} + \mathbf{B}^\top\boldsymbol{\Lambda}_{\phi_t}^2\mathbf{B}).$$

**Theorem 2** (Hessian Bounds under Soft and Hard Constraints)**.** *Under the assumptions above, the curvature of the hard-constrained loss is dominated by bounded scaling factors of $\phi$ and $\phi_t$:*

$$|H_s|_2 \leq \frac{|\mathcal{L}_s|^2_2}{N}|\mathbf{B}|^2_2 + \frac{1}{M}|\mathbf{C}|^2_2, \quad |H_h|_2 \leq \frac{1}{N}(C_1|\mathcal{L}_s|^2_2 + 2C_2|\mathcal{L}_s|_2 + C_3)|\mathbf{B}|^2_2.$$

*where $|\mathcal{L}_s|_2$ is the $L^2$-operator norm of $\mathcal{L}_s$ and*

$$C_1 = \max_{\mathbf{z}} \phi^2(\mathbf{z}), \quad C_2 = \max_{\mathbf{z}} |\phi(\mathbf{z})\phi_t(\mathbf{z})|, \quad \text{and} \quad C_3 = \max_{\mathbf{z}} \phi_t^2(\mathbf{z}).$$

**Remark 2.** *Notice the competition between the PDE residual points and the initial conditions points (reflected in $\mathbf{B}$ and $\mathbf{C}$) in the bound for $H_s$, where for $H_h$, it is all about the operator $\mathcal{L}_s$ and the scaling by $\phi$ and $\phi_t$. The bounds will become much more complex when $\mathcal{P}$ is not linear or $\phi$ also depends on $\mathbf{x}$.*

### 3.3 NTK PERSPECTIVE AND TRAINING DECAY

Finally, we provide a Neural Tangent Kernel (NTK) perspective on how hard constraints would work for reduce the spectral bias and training difficulties of PINNs on solving stiff time-dependent PDEs. To simplify the discussion, we will focus on the PDEs with periodic boundary. Let $\{\mathbf{z}_i = (t_i, \mathbf{x}_i)\}_{i=1}^N \subset (0, T) \times \Omega$ denote collocation (residual) points and $\{\boldsymbol{\zeta}_j = (0, \mathbf{x}_j)\}_{j=1}^M$ denote initial-condition sample points. We define the two residual functions $r_s$ and $r_h$ as

$$r_s(\mathbf{z}_i) = \partial_t u_s(\mathbf{z}_i) + \mathcal{P}[u_s](\mathbf{z}_i) - f(\mathbf{z}_i),$$

and

$$r_h(\mathbf{z}_i) = \phi(\mathbf{z}_i)\partial_t u_h(\mathbf{z}_i) + \mathcal{P}[\psi + \phi u_h](\mathbf{z}_i) + \phi_t(\mathbf{z}_i)u_h(\mathbf{z}_i) - f(\mathbf{z}_i) - \psi_t(\mathbf{z}_i),$$

where $u_s = u_s(t, \mathbf{x}; \boldsymbol{\theta}_s)$ (soft constrained) and $u_h = u_h(t, \mathbf{x}; \boldsymbol{\theta}_h)$ (hard constrained) are the two PINN approximations, which are automatically periodic due to the Fourier layer. The corresponding losses to train $u_s$ and $u_h$ are

$$\text{Loss}_s(u_s(\boldsymbol{\theta}_s)) = \text{Loss}_{IC}(u_s(\boldsymbol{\theta}_s)) + \text{Loss}_{PDE}(u_s(\boldsymbol{\theta}_s))$$

with

$$
\begin{cases}
\text{Loss}_{IC}(u_s(\boldsymbol{\theta}_s)) & = \frac{1}{2M} \sum_{i=1}^{M} |u_s(\boldsymbol{\zeta}_i; \boldsymbol{\theta}_s) - u_0(\mathbf{x}_i^{IC})|^2, \quad \boldsymbol{\zeta}_i = (0, \mathbf{x}_i^{IC}), \\
\text{Loss}_{PDE}(u_s(\boldsymbol{\theta}_s)) & = \frac{1}{2N} \sum_{i=1}^{N} |r_s(\mathbf{z}_i; \boldsymbol{\theta}_s)|^2,
\end{cases}
$$

and $\quad \text{Loss}_h(u_h(\boldsymbol{\theta}_h)) = \frac{1}{2N} \sum_{i=1}^{N} |r_h(\mathbf{z}_i; \boldsymbol{\theta}_h)|^2.$

Both Losses can induce a continuous-time (in terms of $\tau$ not $t$ as $t$ represents the physical time) gradient flow systems as

$$\frac{d\boldsymbol{\theta}_s(\tau)}{d\tau} = -\nabla \text{Loss}_s(u_s(\boldsymbol{\theta}_s(\tau))) \quad \text{and} \quad \frac{d\boldsymbol{\theta}_h(\tau)}{d\tau} = -\nabla \text{Loss}_h(u_h(\boldsymbol{\theta}_s(\tau))),$$

with $\boldsymbol{\theta}_s(0) = \boldsymbol{\theta}_h(0) = \boldsymbol{\theta}_0$. Let the operators $\mathcal{L}_s[u] := \partial_t u + \mathcal{P}[u]$ and $\mathcal{L}_h[u] := \phi \partial_t u + \mathcal{P}[\psi + \phi u] + \phi_t u$. Then with NTK, we obtain

$$
\begin{bmatrix} \frac{du_s(\boldsymbol{\zeta}; \boldsymbol{\theta}_s(\tau))}{d\tau} \\ \frac{d\mathcal{L}_s[u_s](\mathbf{z}; \boldsymbol{\theta}_s(\tau))}{d\tau} \end{bmatrix} = - \begin{bmatrix} \mathbf{K}_{11}^s(\tau) & \mathbf{K}_{12}^s(\tau) \\ \mathbf{K}_{21}^s(\tau) & \mathbf{K}_{22}^s(\tau) \end{bmatrix} \cdot \begin{bmatrix} u_s(\boldsymbol{\zeta}; \boldsymbol{\theta}_s(\tau)) - u_0(\mathbf{x}^{IC}) \\ \mathcal{L}_s[u_s](\mathbf{z}; \boldsymbol{\theta}_s(\tau)) - f(\mathbf{z}) \end{bmatrix}.
$$

with $\mathbf{K}_{11}^s(\tau) \in \mathbb{R}^{M \times M}$, $\mathbf{K}_{12}^s(\tau) \in \mathbb{R}^{M \times N}$ ($\mathbf{K}_{12}^s(\tau) = (\mathbf{K}_{21}^s(\tau))^\top$), and $\mathbf{K}_{22}^s(\tau) \in \mathbb{R}^{N \times N}$ and the entries are given as

$$
\begin{cases}
(\mathbf{K}_{11}^s)_{i,j}(\tau) & = \langle \partial_{\boldsymbol{\theta}} u_s(\boldsymbol{\zeta}_i; \boldsymbol{\theta}_s(\tau)), \partial_{\boldsymbol{\theta}} u_s(\boldsymbol{\zeta}_j; \boldsymbol{\theta}_s(\tau)) \rangle \\
(\mathbf{K}_{12}^s)_{i,j}(\tau) & = \langle \partial_{\boldsymbol{\theta}} u_s(\boldsymbol{\zeta}_i; \boldsymbol{\theta}_s(\tau)), \partial_{\boldsymbol{\theta}} \mathcal{L}_s[u_s](\mathbf{z}_j; \boldsymbol{\theta}_s(\tau)) \rangle \\
(\mathbf{K}_{22}^s)_{i,j}(\tau) & = \langle \partial_{\boldsymbol{\theta}} \mathcal{L}_s[u_s](\mathbf{z}_i; \boldsymbol{\theta}_s(\tau)), \partial_{\boldsymbol{\theta}} \mathcal{L}_s[u_s](\mathbf{z}_j; \boldsymbol{\theta}_s(\tau)) \rangle
\end{cases}
.$$

Meanwhile, for $u_h$, we have

$$\frac{d\mathcal{L}_h[u_h](\mathbf{z}; \boldsymbol{\theta}_h(\tau))}{d\tau} = -\mathbf{K}^h(\mathcal{L}_h[u_h](\mathbf{z}; \boldsymbol{\theta}_h(\tau)) - f(\mathbf{z}) - \psi_t(\mathbf{z})), \quad \mathbf{K}^h \in \mathbb{R}^{M \times M},$$

with the entries of $\mathbf{K}^h$ being

$$(\mathbf{K}^h)_{i,j}(\tau) = \langle \partial_{\boldsymbol{\theta}} \mathcal{L}_h[u_h](\mathbf{z}_i; \boldsymbol{\theta}_h(\tau)), \partial_{\boldsymbol{\theta}} \mathcal{L}_h[u_h](\mathbf{z}_j; \boldsymbol{\theta}_h(\tau)) \rangle.$$

Setting $\mathbf{K}^s = \begin{bmatrix} \mathbf{K}_{11}^s & \mathbf{K}_{12}^s \\ \mathbf{K}_{21}^s & \mathbf{K}_{22}^s \end{bmatrix}$, and assuming $\mathbf{K}^s(\tau) \approx \mathbf{K}^s(0) = \mathbf{K}_*^s$ and $\mathbf{K}^h(\tau) \approx \mathbf{K}^h(0) = \mathbf{K}_*^h$, then the decays satisfy

$$
\begin{bmatrix} \frac{du_s(\boldsymbol{\zeta}; \boldsymbol{\theta}_s(\tau))}{d\tau} \\ \frac{d\mathcal{L}_s[u_s](\mathbf{z}; \boldsymbol{\theta}_s(\tau))}{d\tau} \end{bmatrix} - \begin{bmatrix} u_0(\mathbf{x}^{IC}) \\ f(\mathbf{z}) \end{bmatrix} = -e^{-\mathbf{K}_*^s \tau} \begin{bmatrix} u_0(\mathbf{x}^{IC}) \\ f(\mathbf{z}) \end{bmatrix} = -\mathbf{Q}_s^\top e^{-\boldsymbol{\Lambda}_s \tau} \mathbf{Q}_s \begin{bmatrix} u_0(\mathbf{x}^{IC}) \\ f(\mathbf{z}) \end{bmatrix},
$$

where $\mathbf{K}_*^s = \mathbf{Q}_s^\top \boldsymbol{\Lambda}_s \mathbf{Q}_s$; moreover

$$\frac{d\mathcal{L}_h[u_h](\mathbf{z}; \boldsymbol{\theta}_h(\tau))}{d\tau} - (f(\mathbf{z}) + \psi_t(\mathbf{z})) = -e^{\mathbf{K}_*^h \tau}(f(\mathbf{z}) + \psi_t(\mathbf{z})) = -\mathbf{Q}_h^\top e^{-\boldsymbol{\Lambda}_h \tau} \mathbf{Q}(f(\mathbf{z}) + \psi_t(\mathbf{z})),$$

where $\mathbf{K}_*^h = \mathbf{Q}_h^\top \boldsymbol{\Lambda}_h \mathbf{Q}_h$. One can see that the training decay of not using the hard-constrained transformation depends on both the eigenvalues of $\mathbf{K}_{11}^s$ and $\mathbf{K}_{22}^s$; whereas the training decay of the hard-constrained transformation depends only on $\mathbf{K}^h$, hence reducing the spectral bias.

## 4 EXAMPLES

In this section, we examine the hard-constrained PINN (HC-PINN) on a diverse set of prototypical stiff, time-dependent PDEs with periodic boundary conditions, aiming to demonstrate its robustness and flexibility in addressing a broad class of challenging problems. These equations often involve strong nonlinearities, stiffness, and parameter sensitivities, posing significant challenges for data-driven methods such as PINNs. Nonetheless, our results illustrate the capacity of the proposed

framework to overcome these difficulties and accurately approximate the solutions across these varied regimes.

To begin, we employ Latin hypercube sampling to generate collocation points across the full spatial-temporal domain, including its boundary. To facilitate training and better incorporate the initial condition into the solution structure, we adopt the following transformation:

$$\tilde{u}(t, \mathbf{x}) = e^{-Ct} * u_0(\mathbf{x}) + t * u_{nn}(t, \mathbf{x}), \quad \psi = e^{-Ct} * u_0(\mathbf{x}), \phi = t, \quad (t, \mathbf{x}) \in [0, T] \times \bar{\Omega},$$

where $u_0(\mathbf{x})$ denotes the initial condition, and we test two values of $C \in \{0.1, 1\}$, to assess how the decay rate affects learned dynamics. The transformed PINN $u_{nn}$ is trained to minimize a loss function that accounts for the PDE residuals, which now includes IC/BC.

We present the results in four major categories, $(I)$ ablation study across 7 different stiff PDEs where stiffness comes in different forms, such as sharp phase transition in Allen-Cahn, higher-order derivatives (forth order) in Cahn-Hilliard and Kuramoto-Sivashinsky, coupled outputs form Gray-Scott and Shrödinger systems, and high frequency modes in fast moving linear transport; $(II)$ comparison to other start-of-the-art PINNs such as the Causal PINN (Wang et al., 2022) and Enhanced RBA PINN (Anagnostopoulos et al., 2024) on Allen-Cahn, Cahn-Hilliard and Kuramoto-Sivashinsky; $(III)$ application of HC-PINNs on a $2D$ Allen-Cahn; $(IV)$ sensitivity analysis on the parameters $m$ for the Allen-Cahn equation with two different initial conditions.

|  | Causal training (MLP) | Enhanced RBA | HC-PINNs |
|---|---|---|---|
| AC (Case I) | $6.95 \times 10^{-2}$ | $2.62 \times 10^{-3}$ | $8.29 \times 10^{-4}$ |
| AC (Case II) | $1.78 \times 10^{-2}$ | $1.08 \times 10^{-3}$ | $2.14 \times 10^{-4}$ |
| CH | $3.49 \times 10^{-1}$ | $9.83 \times 10^{-2}$ | $5.61 \times 10^{-4}$ |
| KS | $2.72 \times 10^{-1}$ | $3.64 \times 10^{-2}$ | $5.37 \times 10^{-4}$ |

Table 1: Relative $L^2$ errors for the Allen–Cahn (AC), Cahn-Hilliard (CH) and Kuramoto–Sivashinsky (KS) equations obtained after $50k$ training iterations. All methods are implemented under the same $50k$ iteration budget for a fair comparison. Note that the reported errors for Causal training (Wang et al., 2022) and Enhanced RBA (Anagnostopoulos et al., 2024) differ from those in their original papers, where results were obtained with $300k$ iterations.

Figure 1 provides a summary of our ablation study over 7 different benchmarks; it shows the comparison between the reference exact solutions and our best HC-PINN predictions, together with the corresponding absolute errors; the benchmarks include Allen–Cahn, Cahn–Hilliard, Kuramoto–Sivashinsky, Gray–Scott, nonlinear Schrödinger, and linear advection equations (with different speeds $c = 50, 60$). Complementing these visual comparisons, Table 1 summarizes the comparison study between HC-PINN and two other PINN variants in terms of relative $L^2$ errors for Allen–Cahn, Cahn-Hilliard and Kuramoto–Sivashinsky equations under a strictly controlled setting: all methods are trained for $50k$ iterations, same data and parameter settings to ensure fairness. Notably, while prior works originally reported results with $300k$ iterations, HC-PINNs achieve substantially lower errors even with a much shorter training budget. Overall, these extensive experiments highlight the effectiveness and adaptability of our PINN scheme in solving a wide variety of stiff time-dependent PDEs. The results provide not only strong empirical validation but also a reliable and reproducible benchmark for future developments in this area. For full experimental details and visualizations, see Section D in the appendix.

The results of solving the $2D$ Allen–Cahn equation in Figure 2 were obtained after training the network for $50k$ iterations using the initial condition $u(0, x, y) = \sin(4\pi x)\cos(4\pi y)$, with $\gamma_1 = 0.0001$ and $\gamma_2 = 1$ in Eq. 7, and periodic boundary conditions. The HC-PINN accurately captures the solution evolution, yielding exceptionally small relative $L^2$ errors: $8.04 \times 10^{-5}$ at $t = 0.5$ and $1.33 \times 10^{-4}$ at $t = 1$. This demonstrates the high accuracy and stability of our method for 2D Allen–Cahn dynamics. At last, we perform a sensitivity study on the number of Fourier modes $m$ for the Allen–Cahn benchmarks to assess how this key hyperparameter influences the accuracy of HC-PINNs. The results demonstrate that accuracy is highly dependent on spectral resolution: while small $m$ yields stable and low errors, performance deteriorates significantly once $m$ exceeds a critical threshold.

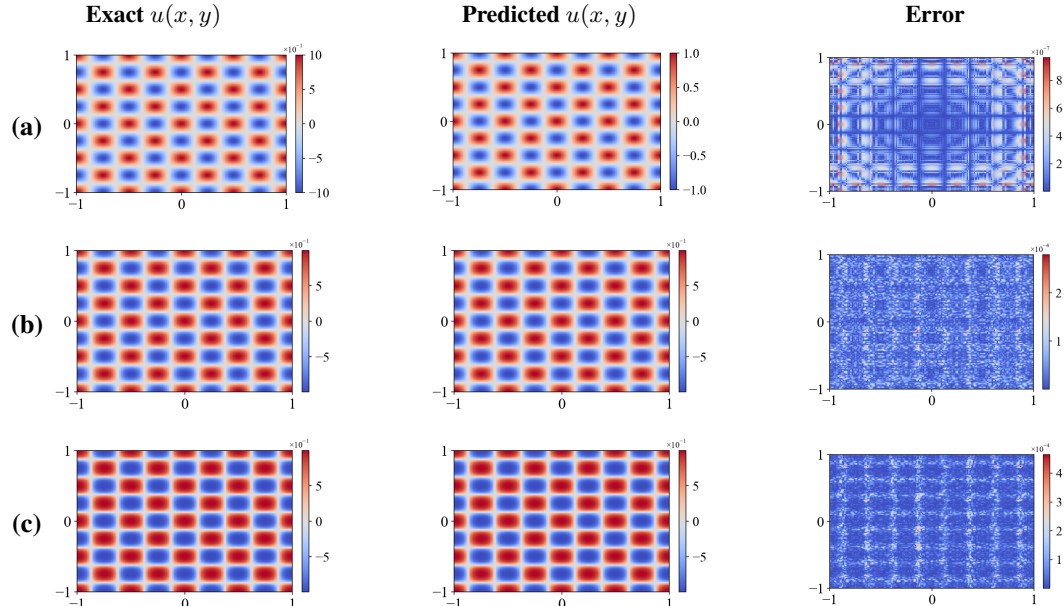

Figure 2: Reference, HC-PINN predicted solution, and absolute error of the 2D Allen Cahn equation at different time snapshots (a) $t = 0$ , (b) $t = 0.5$ , (c) $t = 1$.

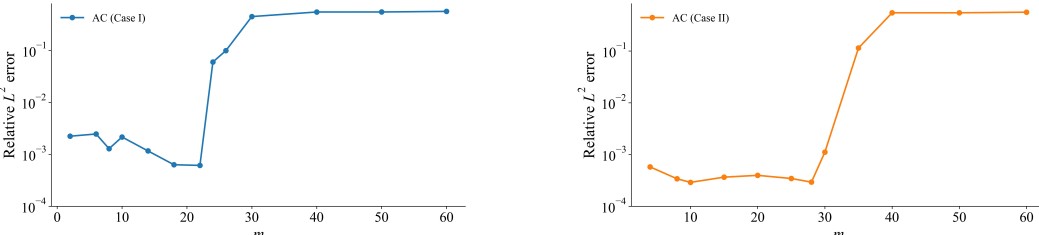

Figure 3: Sensitivity of HC-PINNs to the number of Fourier modes $m$ in the Allen–Cahn equations. The plot shows the relative $L^2$ error as a function of $m$.

## 5 CONCLUSION

We provide a comprehensive study of hard-constraints and self-adaptive loss weights to study the stabilization of PINN training on solving stiff time-dependent PDEs. Our comprehensive study includes an ablation study of seven distinct types of PDES, comparison study, a theoretical analysis, an application to $2D$ Allen Chan, and a sensitive analysis of $m$ (the number of Fourier modes).

From the ablation study, we found out that the transformation provided by hard constraint significantly alleviates the training challenges typically associated with stiff time-dependent PDEs. Our ablation study demonstrates effectiveness across seven distinct types of such PDEs, encompassing scalar-valued and vector-valued (including complex-valued) outputs. This approach not only enhances training efficiency but also improves the accuracy and stability of the learned solutions, particularly in the presence of stiffness, where conventional PINNs often struggle. We provide theoretical insights using NTK into how hard-constrained transformation is reducing the training difficulties, especially in terms of reducing the spectral bias. The comparison study focuses on comparing our HC-PINN to two other state-of-the-art PINN variants on established examples.

Future work includes extending the hard-cosntraint framework to more two- and three-dimensional PDEs, which pose additional challenges in both architecture design and computational cost. Ongoing research also focuses on identifying optimal transformations $\psi$ and $\phi$, which govern the embedding of IC and BC into the network structure, with the goal of further enhancing performance and generalizability.

## REPRODUCIBILITY STATEMENT

We have made substantial efforts to ensure the reproducibility of our work. All methodological details, including the hard-constraint framework, loss formulations, and training strategies such as mini-batching and self-adaptive loss weights, are clearly described in Section 2 and Appendix C. Theoretical assumptions and complete proofs of our results are provided in Appendix C.4, ensuring transparency of the analysis. For empirical validation, we conducted extensive ablation studies on seven benchmark stiff PDEs, detailed in Section 4 and Appendix D, with standardized settings (e.g., fixed training iterations, same collocation points, and data sampling strategies) to allow fair comparison and replication. We also report sensitivity analyses on key hyperparameters (e.g., Fourier modes $m$, decay rate $C$), presented in Section 4 and ablation studies in Appendix D, to document the robustness of our approach. Furthermore, all datasets used in experiments are synthetic PDE solutions, and we describe the data generation and preprocessing steps in the appendix. Finally, we release our code with full implementations of HC-PINNs, training scripts, and data generation procedures to facilitate independent verification of our results in supplementary materials.

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

## A    Ethics Statement

This work focuses on the development of training strategies for physics-informed neural networks (PINNs) to solve stiff time-dependent partial differential equations (PDEs). All datasets used in this study are synthetically generated from well-established PDE models and do not involve human subjects, personal data, or sensitive information. Our methods are intended for advancing scientific machine learning and have no foreseeable direct harmful applications. We have carefully documented all experimental settings and theoretical assumptions to ensure research integrity and reproducibility. We declare that there are no conflicts of interest or external sponsorships that could have influenced the results.

## B    The Use of Large Language Models (LLMs)

In preparing this manuscript, large language models (LLMs) were used only for grammar checking and minor language polishing. No part of the research ideation, technical content, analysis, or scientific writing relied on LLMs. The authors take full responsibility for the content of the paper.

## C    Methodology: Additional Details

In this section, we discuss the additional methods used in the ablation study, including mini-batching and self-adaptive loss weights.

### C.1    Mini-batching

We have eliminated the IC/BC losses, but the transformed PDE loss might create additional difficulties in training due to the introduction of IC/BC functions into the PDE. We adopt a machine learning technique, called mini-batching, commonly used to reduce training complexity by evaluating the loss on randomly selected subsets of collocation data. In PINNs, this reduces memory costs, especially when handling large numbers of collocation points. Sampling mini-batches from collocation points also helps stabilize and accelerate optimization. So we consider mini-batching in the training for the PDE residual loss. As pointed out in (Wight & Zhao, 2020), the mini-batching technique is similar to time-marching resampling but without an explicit time grid.

### C.2    Self-Adaptive PINNs

The proper choice of penalty term $\lambda$ can be updated from derived formula as mentioned in (Wang et al., 2020b). However, data-driven discovery of $\lambda$ is possible. Outline in (McClenny & Braga-Neto,

2020), the loss is further modified as

$$\text{Data Loss}(\{\lambda_i^{IC}\}_{i=1}^{N_{IC}}, \{\lambda_i^{BC}\}_{i=1}^{N_{BC}})$$

$$= \frac{1}{N_{IC}} \sum_{i=1}^{N_{IC}} F_{mask}(\lambda_i^{IC}) |u_{nn}(0, \mathbf{x}_i^{IC}) - u_0(\mathbf{x}_i^{IC})|^2$$

$$+ \sum_{i=1}^{d} \frac{1}{N_{BC}} \sum_{j=1}^{N_{BC}} F_{mask}(\lambda_j^{BC}) |u_{nn}(t_j^{BC}, \mathbf{x}_j^{BC}) - u_{nn}(t_j^{BC}, \mathbf{x}_j^{BC} + P\mathbf{e}_i)|^2,$$

and

$$\text{Physical Loss}(\{\lambda_i^{CL}\}_{i=1}^{N_{CL}}) = \frac{1}{N_{CL}} \sum_{i=1}^{N_{CL}} F_{mask}(\lambda_i^{CL}) |(\frac{\partial u}{\partial t} - \mathcal{P}(u_{nn}) - f)(t_i^{CL}, \mathbf{x}_i^{CL})|^2.$$

The set of loss weights, $\{\lambda_i^{IC}\}_{i=1}^{N_{IC}} \cup \{\lambda_i^{BC}\}_{i=1}^{N_{BC}} \cup \{\lambda_i^{CL}\}_{i=1}^{N_{CL}}$, are designed in a way to increase when needed during any training epoch in order to combat the spectral bias. Adaptive methods are essential to ensure that the neural network accurately addresses the challenging spots in the solutions of "stiff" PDEs. In our experiments, we employ self-adaptive PINNs as proposed by (McClenny & Braga-Neto, 2020) to train PINNs adaptively. This approach involves fully trainable adaptation weights applied individually to each training point, allowing the neural network to independently identify and focus on the challenging regions of the solution.

### C.3 PERFORMANCE MEASURES

Let $\{x_i, t_i\}_{i=1}^{N}$ be a collection of $N$ data points where we evaluate both the reference solution $u(x, t)$ and the neural network output $\mathcal{U}(x, t)$. To measure the accuracy of the trained model, we compute the relative $L_1$ norm $\mathcal{E}_1$ and relative $L_2$ norm $\mathcal{E}_2$, defined by:

$$\mathcal{E}_1 = \frac{\sum_{i=1}^{N} |\mathcal{U}(x_i, t_i) - u(x_i, t_i)|}{\sum_{i=1}^{N} |u(x_i, t_i)|}, \quad \mathcal{E}_2 = \frac{\sqrt{\sum_{i=1}^{N} |\mathcal{U}(x_i, t_i) - u(x_i, t_i)|^2}}{\sqrt{\sum_{i=1}^{N} |u(x_i, t_i)|^2}}.$$

These metrics measure the discrepancy between the predicted and reference solutions in average (energy-based) and absolute (mass-based) senses, respectively. The relative form normalizes the error with respect to the reference solution, enabling fair comparisons across different scales and variables. To rigorously assess the accuracy of our results, we implemented a semi-implicit spectral method in Matlab. This allows us to make precise comparisons between the ground-truth solutions $u(x_i, t_i)$ and the PINN-generated solutions $\mathcal{U}(x_i, t_i)$.

### C.4 PROOFS OF THE THEOREMS

We present the proofs for the two aforementioned theorem.

**Theorem** (Well Conditioning). *The transformation is well defined when $\phi$ and $\psi$ is $C^1$ in time and have $K^{th}$ order partial derivatives w.r.t $\mathbf{x}$. Moreover, $\phi_t(0, \mathbf{x}) \neq 0$. Then $u_{nn}$ will satisfy a new PDE*

$$\begin{cases} \phi \partial_t u_{nn} + \mathcal{P}[\psi + \phi u_{nn}] + \phi_t u_{nn} = f - \psi_t, & (t, \mathbf{x}) \in (0, T] \times \Omega, \\ u_{nn}(0, \mathbf{x}) = \frac{u_t(0, \mathbf{x}) - \psi_t(0, \mathbf{x})}{\phi_t(0, \mathbf{x})}, & \mathbf{x} \in \bar{\Omega}. \end{cases}$$

*Proof.* Since $\mathcal{L}_s[u] = \partial_t u + \mathcal{P}[u]$ is applied to the transformation $u = \psi + \phi u_{nn}$ and $u_{nn}$ is constructed with hyper tangent as the activation function, we only need $\psi$ and $\phi$ to satisfy the required regularity from $\mathcal{L}_s$ so that the operator can still be applied, hence $\psi$ and $\phi$ have to have first-order time derivative and $K^{th}$-order partial derivatives w.r.t to $\mathbf{x}$, we further require them to be continuous in order to allow classical solutions to exist for $\mathcal{L}_s$. Meanwhile from the transformation we have

$$u_{nn}(t, \mathbf{x}) = \frac{u(t, \mathbf{x}) - \psi(t, \mathbf{x})}{\phi(t, \mathbf{x})}, \quad \text{for } (t, \mathbf{x}) \in (0, T] \times \Omega.$$

As $t \to 0$, we have $\psi(0, \mathbf{x}) = u_0(\mathbf{x})$ and $\phi(0, \mathbf{x}) = 0$, hence by L'Hôpital's rule

$$\lim_{t \to 0} u_{nn}(t, \mathbf{x}) = \lim_{t \to 0} \frac{u(t, \mathbf{x}) - \psi(t, \mathbf{x})}{\phi(t, \mathbf{x})} = \lim_{t \to 0} \frac{u_t(t, \mathbf{x}) - \psi_t(t, \mathbf{x})}{\phi_t(t, \mathbf{x})},$$

when $\phi_t(0, \mathbf{x}) \neq 0$, then limit is defined and we can have

$$u_{nn}(0, \mathbf{x}) = \lim_{t \to 0} u_{nn}(t, \mathbf{x}) = \frac{u_t(0, \mathbf{x}) - \psi_t(0, \mathbf{x})}{\phi_t(0, \mathbf{x})}.$$

Moreover

$$\mathcal{L}_s(\psi + \phi u_{nn}) = \partial_t(\psi + \phi u_{nn}) + \mathcal{P}[\psi + \phi u_{nn}] = \phi \partial_t u_{nn} + \mathcal{P}[\psi + \phi u_{nn}] + \phi_t u_{nn} + \psi_t,$$

hence the new PDE for $u_{nn}$. $\qquad\qquad\square$

Next, the theorem on Hessian bounds.

**Theorem.** *Assume that $\mathcal{P}$ is linear and $\phi(t, \mathbf{x}) = \phi(t) \neq 0$ for $t > 0$, then for a one-hidden layer network, we have for a network $u(\mathbf{y}; \theta)$ with $\boldsymbol{\theta} \in \mathbb{R}^D$ being the network parameters, the following relationship is satisfied*

$$u(\mathbf{y}; \boldsymbol{\theta}) = \mathbf{h}^\top(\mathbf{y})\boldsymbol{\theta}, \quad \mathbf{h} = [h_1(\mathbf{y}) \quad \cdots \quad h_D(\mathbf{y})]^\top,$$

*where $\mathbf{y} = \mathbf{z} = (t, \mathbf{x}) \in (0, T] \times \Omega$ or $\mathbf{y} = \boldsymbol{\zeta} = (0, \mathbf{x}^{IC}) \in \{0\} \times \bar{\Omega}$. Therefore the two losses become*

$$Loss_s(u(\mathbf{y}; \boldsymbol{\theta})) = \frac{1}{2N}|\mathbf{A}\boldsymbol{\theta} - \mathbf{f}|^2_{\ell^2(\mathbb{R}^N)} + \frac{1}{2M}|\mathbf{C}\boldsymbol{\theta} - \mathbf{u}_0|^2_{\ell^2(\mathbb{R}^M)}, \quad \text{(soft constrained)}$$

$$Loss_h(u(\mathbf{z}; \boldsymbol{\theta})) = \frac{1}{2N}|\boldsymbol{\Lambda}_\phi \mathbf{A}\boldsymbol{\theta} + \boldsymbol{\Lambda}_{\phi_t}\mathbf{B}\boldsymbol{\theta} - \tilde{\mathbf{f}}|^2_{\ell^2(\mathbb{R}^N)}, \quad \text{(hard constrained)}$$

*where*

$$\mathbf{A} = \begin{bmatrix} \mathcal{L}_s[\mathbf{h}^\top(\mathbf{z}_1)] \\ \vdots \\ \mathcal{L}_s[\mathbf{h}^\top(\mathbf{z}_N)] \end{bmatrix}, \quad \mathbf{B} = \begin{bmatrix} \mathbf{h}^\top(\mathbf{z}_1) \\ \vdots \\ \mathbf{h}^\top(\mathbf{z}_N) \end{bmatrix}, \quad \text{and} \quad \mathbf{C} = \begin{bmatrix} \mathbf{h}^\top(\boldsymbol{\zeta}_1) \\ \vdots \\ \mathbf{h}^\top(\boldsymbol{\zeta}_M) \end{bmatrix},$$

*where $\mathbf{x}_i^{IC} \in \bar{\Omega}$; and for the diagonal matrices $\boldsymbol{\Lambda}_\phi$ and $\boldsymbol{\Lambda}_{\phi_t}$*

$$\boldsymbol{\Lambda}_\phi = diag(\phi(\mathbf{z}_1), \cdots, \phi(\mathbf{z}_N)) \quad \text{and} \quad \boldsymbol{\Lambda}_{\phi_t} = diag(\phi_t(\mathbf{z}_1), \cdots, \phi_t(\mathbf{z}_N)),$$

*and the vectors*

$$\mathbf{f} = \begin{bmatrix} f(\mathbf{z}_1) \\ \vdots \\ f(\mathbf{z}_N) \end{bmatrix}, \quad \tilde{\mathbf{f}} = \begin{bmatrix} f(\mathbf{z}_1) - \mathcal{L}_s[\psi](\mathbf{z}_1) \\ \vdots \\ f(\mathbf{z}_N) - \mathcal{L}_s[\psi](\mathbf{z}_N) \end{bmatrix}, \quad \text{and} \quad \mathbf{u}_0 = \begin{bmatrix} u_0(\mathbf{x}_1^{IC}) \\ \vdots \\ u_0(\mathbf{x}_M^{IC}) \end{bmatrix}.$$

*Therefore the Hessians of the losses are*

$$H_s(\boldsymbol{\theta}) = \frac{1}{N}\mathbf{A}^\top\mathbf{A} + \frac{1}{M}\mathbf{C}^\top\mathbf{C},$$

$$H_h(\boldsymbol{\theta}) = \frac{1}{N}(\mathbf{A}^\top\boldsymbol{\Lambda}_\phi^2\mathbf{A} + 2\mathbf{A}^\top\boldsymbol{\Lambda}_\phi\boldsymbol{\Lambda}_{\phi_t}\mathbf{B} + \mathbf{B}^\top\boldsymbol{\Lambda}_{\phi_t}^2\mathbf{B}),$$

*Then the Hessians are bounded as*

$$|H_s|_2 \leq \frac{|\mathcal{L}_s|^2_2}{N}|\mathbf{B}|^2_2 + \frac{1}{M}|\mathbf{C}|^2_2,$$

$$|H_h|_2 \leq \frac{1}{N}(C_1|\mathcal{L}_s|^2_2 + 2C_2|\mathcal{L}_s|_2 + C_3)|\mathbf{B}|^2_2.$$

*where $|\mathcal{L}_s|_2$ is the $L^2$-operator norm of $\mathcal{L}_s$ and*

$$C_1 = \max_{\mathbf{z}} \phi^2(\mathbf{z}), \quad C_2 = \max_{\mathbf{z}} |\phi(\mathbf{z})\phi_t(\mathbf{z})|, \quad \text{and} \quad C_3 = \max_{\mathbf{z}} \phi_t^2(\mathbf{z}).$$

*Proof.* Once we have $u(\mathbf{y}; \boldsymbol{\theta}) = \mathbf{h}^\top(\mathbf{y})\boldsymbol{\theta}$, then the loss from the soft-constrained PINN can be re-written as

$$\text{Loss}_s(u(\boldsymbol{\theta})) = \frac{1}{2N}\sum_{i=1}^{N}|(\mathcal{L}_s[u(\cdot; \boldsymbol{\theta})] - f)(\mathbf{z}_i)|^2 + \frac{1}{2M}\sum_{j=1}^{M}|u(\boldsymbol{\zeta}_i; \boldsymbol{\theta}) - u_0(\mathbf{x}_i^{IC})|^2,$$

If we define the matrices

$$\mathbf{A} = \begin{bmatrix} \mathcal{L}_s[\mathbf{h}^\top(\mathbf{z}_1)] \\ \vdots \\ \mathcal{L}_s[\mathbf{h}^\top(\mathbf{z}_N)] \end{bmatrix} \quad \text{and} \quad \mathbf{C} = \begin{bmatrix} \mathbf{h}^\top(\boldsymbol{\zeta}_1) \\ \vdots \\ \mathbf{h}^\top(\boldsymbol{\zeta}_M) \end{bmatrix},$$

and the vectors

$$\mathbf{f} = \begin{bmatrix} f(\mathbf{z}_1) \\ \vdots \\ f(\mathbf{z}_N) \end{bmatrix} \quad \text{and} \quad \mathbf{u}_0 = \begin{bmatrix} u_0(\mathbf{x}_1^{IC}) \\ \vdots \\ u_0(\mathbf{x}_M^{IC}) \end{bmatrix}.$$

then $\text{Loss}_s(u(\boldsymbol{\theta})) = \frac{1}{2N}|\mathbf{A}\boldsymbol{\theta} - \mathbf{f}|_{\ell^2(\mathbb{R}^N)}^2 + \frac{1}{2M}|\mathbf{C}\boldsymbol{\theta} - \mathbf{u}_0|_{\ell^2(\mathbb{R}^M)}^2$. Similarly, for the loss

$$\text{Loss}_h(u(\boldsymbol{\theta})) = \frac{1}{2N}\sum_{i=1}^{N}|(\mathcal{L}_s[\psi(\cdot) + \phi(\cdot)u(\cdot; \boldsymbol{\theta})] - f)(\mathbf{z}_i)|^2,$$

Since

$$\mathcal{L}_s[\psi + \phi u] = \phi(u_t + \mathcal{P}[u]) + \phi_t u + \psi_t, \quad \mathcal{P} \text{ is linear and } \phi(t, \mathbf{x}) = \phi(t).$$

Hence with

$$\mathbf{B} = \begin{bmatrix} \mathbf{h}^\top(\mathbf{z}_1) \\ \vdots \\ \mathbf{h}^\top(\mathbf{z}_N) \end{bmatrix} \quad \text{and} \quad \tilde{\mathbf{f}} = \begin{bmatrix} f(\mathbf{z}_1) - \mathcal{L}_s[\psi](\mathbf{z}_1) \\ \vdots \\ f(\mathbf{z}_N) - \mathcal{L}_s[\psi](\mathbf{z}_N) \end{bmatrix},$$

and the two diagonal matrices

$$\boldsymbol{\Lambda}_\phi = \text{diag}(\phi(\mathbf{z}_1), \cdots, \phi(\mathbf{z}_N)) \quad \text{and} \quad \boldsymbol{\Lambda}_{\phi_t} = \text{diag}(\phi_t(\mathbf{z}_1), \cdots, \phi_t(\mathbf{z}_N)),$$

we have $\text{Loss}_h(u(\boldsymbol{\theta})) = \frac{1}{2N}|\boldsymbol{\Lambda}_\phi\mathbf{A}\boldsymbol{\theta} + \boldsymbol{\Lambda}_{\phi_t}\mathbf{B}\boldsymbol{\theta} - \tilde{\mathbf{f}}|_{\ell^2(\mathbb{R}^N)}^2$. Then the two gradients are

$$\nabla_{\boldsymbol{\theta}}\text{Loss}_s(u(\boldsymbol{\theta})) = \frac{1}{N}(\mathbf{A}^\top\mathbf{A}\boldsymbol{\theta} - \mathbf{A}^\top\mathbf{f}) + \frac{1}{M}(\mathbf{C}^\top\mathbf{C}\boldsymbol{\theta} - \mathbf{C}^\top\mathbf{u}_0),$$

$$\nabla_{\boldsymbol{\theta}}\text{Loss}_h(u(\boldsymbol{\theta})) = \frac{1}{N}((\boldsymbol{\Lambda}_\phi\mathbf{A} + \boldsymbol{\Lambda}_{\phi_t}\mathbf{B})^\top(\boldsymbol{\Lambda}_\phi\mathbf{A} + \boldsymbol{\Lambda}_{\phi_t}\mathbf{B})\boldsymbol{\theta} - (\boldsymbol{\Lambda}_\phi\mathbf{A} + \boldsymbol{\Lambda}_{\phi_t}\mathbf{B})^\top\tilde{\mathbf{f}}).$$

Differentiating w.r.t to $\boldsymbol{\theta}$ again, we obtain the Hessians as

$$H_s(\boldsymbol{\theta}) = \frac{1}{N}\mathbf{A}^\top\mathbf{A} + \frac{1}{M}\mathbf{C}^\top\mathbf{C},$$

$$H_h(\boldsymbol{\theta}) = \frac{1}{N}(\mathbf{A}^\top\boldsymbol{\Lambda}_\phi^2\mathbf{A} + 2\mathbf{A}^\top\boldsymbol{\Lambda}_\phi\boldsymbol{\Lambda}_{\phi_t}\mathbf{B} + \mathbf{B}^\top\boldsymbol{\Lambda}_{\phi_t}^2\mathbf{B}),$$

Recall that $\mathcal{L}_s[u] = u_t + \mathcal{P}[u]$, hence by the property of an $L^2$ function/operator norm on $\mathcal{L}_s$, we have

$$|\mathcal{L}_s[u]|_2 \leq |\mathcal{L}_s|_2|u|_2, \quad \text{This } |\cdot|_2 \text{ is a norm on functions.}$$

Then we have $|\mathbf{A}^\top\mathbf{A}|_2 \leq |\mathcal{L}_s|_2^2|\mathbf{B}|_2^2$ (Here $|\mathbf{A}|_2$ is a matrix norm). And if we define

$$C_1 = \max_{\mathbf{z}} \phi^2(\mathbf{z}), \quad C_2 = \max_{\mathbf{z}} |\phi(\mathbf{z})\phi_t(\mathbf{z})|, \quad \text{and} \quad C_3 = \max_{\mathbf{z}} \phi_t^2(\mathbf{z}).$$

We can have

$$|\mathbf{A}^\top\boldsymbol{\Lambda}_\phi^2\mathbf{A}|_2 \leq C_1|\mathbf{A}^\top\mathbf{A}|_2 \leq |\mathcal{L}_s|_2^2|\mathbf{B}|_2^2,$$

$$|\mathbf{A}^\top\boldsymbol{\Lambda}_\phi\boldsymbol{\Lambda}_{\phi_t}\mathbf{A}|_2 \leq C_2|\mathbf{A}^\top\mathbf{B}| \leq C_2|\mathcal{L}_s|_2|\mathbf{B}|_2^2,$$

$$|\mathbf{B}^\top\boldsymbol{\Lambda}_{\phi_t}\mathbf{B}|_2 \leq C_3|\mathbf{B}^\top\mathbf{B}|_2 \leq C_3|\mathbf{B}|_2^2$$

Putting them back in, we obtain the final bounds

$$|H_s|_2 \leq \frac{|\mathcal{L}_s|_2^2}{N}|\mathbf{B}|_2^2 + \frac{1}{M}|\mathbf{C}|_2^2,$$

$$|H_h|_2 \leq \frac{1}{N}(C_1|\mathcal{L}_s|_2^2 + 2C_2|\mathcal{L}_s|_2 + C_3)|\mathbf{B}|_2^2.$$

$\square$

## D   EXAMPLES: ADDITIONAL DETAILS

We provide additional details for the ablation study, comparison study, $2D$ application and sensitivity study of $m$ here. We also explain why each PDE or PDE system is chosen due to their particular difficulties for solvers in each sub-section.

### D.1   IMPLEMENTATION DETAILS

All experiments were implemented in PyTorch ($v2.2$) and executed on both CUDA-enabled NVIDIA GPUs and Apple Silicon devices via the MPS backend. Our code runs efficiently across Linux (Ubuntu 22.04), macOS (Sonoma), and Windows 11 platforms. Primary training was conducted on an NVIDIA $A100$ GPU (40 GB memory) and a MacBook Pro with Apple M1 Pro chip (16-core GPU, 32 GB unified memory).

For each benchmark, we used a fully connected neural network with 7 hidden layers, each containing 32 neurons and tanh activation functions, to represent the latent solution functions. The same initial weights were used across all ablation experiments for consistency.

Model training was carried out using a two-stage strategy: first, we trained using the Adam optimizer with an initial learning rate of $10^{-3}$ for $50,000$ steps. This was followed by fine-tuning with the L-BFGS-B optimizer using a learning rate of 1 for $5,000$ steps. We did not perform hyperparameter sweeps, and the total number of training epochs was fixed across all experiments.

This paper does not rely on any datasets. We used Latin Hypercube Sampling (LHS) to generate a fixed set of $16,384$ interior collocation points, along with $128$ initial data points and $128$ boundary data points. This same sampling configuration was used consistently across all experiments to ensure comparability. For PDE systems, we employed a mini-batching strategy with 5 batches of size $3,280$ to improve memory efficiency during training.

Ablation studies were conducted by disabling one component of our method at a time, while keeping all other settings fixed, to evaluate its contribution using relative $L^2$ and relative $L^1$ error metrics, which measure the discrepancy between the predicted and reference solutions in average (energy-based) and absolute (mass-based) senses, respectively. The relative form normalizes the error with respect to the reference solution, enabling fair comparisons across different scales and variables.

We also compare HC-PINN with causal training (MLP) and enhanced RBA using the same data and a total of 50k iterations, instead of the 300k iterations used in the original works. For the sensitivity study, experiments with different $m$ values are conducted on the same points, using the same number of iterations and the same architecture. For 2D training, we use $4,096$ initial collocation points, $10,000$ residual collocation points per segment, and $800$ boundary points per segment. The architecture is the same as in the 1D training.

### D.2   ALLEN-CAHN PDE

The Allen-Cahn type PDE is a classical phase-field model that has been widely used to study phase separation phenomena Bazant (2017). It has numerous practical applications across a range of fields, including materials science Allen & Cahn (1979); Shen & Yang (2010), biological systems Hyman et al. (2014); Takatori & Brady (2015), and electrochemical systems Horstmann et al. (2013); Tian et al. (2015). We begin with the 1-dim Allen-Cahn equation with periodic boundary conditions formulated as follows:

$$u_t - \gamma_1 u_{xx} + \gamma_2(u^3 - u) = 0, \ (t,x) \in (0,T] \times (a,b),$$
$$u(0,x) = u_0(x), \ x \in [a,b], \quad u(t,a) = u(t,b), \ t \in [0,T], \tag{7}$$

where $\gamma_1, \gamma_2 > 0, T > 0, a < b$ are prescribed constants. As $\gamma_2$ increases, the transition interface of the solutions is sharper, which makes it harder to solve the AC equation numerically.

#### D.2.1   CASE $I$

We first tested on Allen-Cahn equation with the following, $u_0(x) = x^2 \cos(\pi x)$, $T = 1$, $a = -1$, $b = 1$, $\gamma_1 = 0.001$, and $\gamma_2 = 5$.

For this example, we first test the baseline PINN approach Raissi et al. (2019). As shown in Table 2, the baseline PINN alone fails to accurately solve the Allen-Cahn equation, motivating the development of a more effective training pipeline. To address this, we employ all proposed techniques (except mini-batching) and conduct an ablation study by disabling individual components one at a time. Additionally, we introduce two different decay functions for the initial condition, $e^{-1t}$ and $e^{-0.1t}$, to investigate their influence on the solution. The results are summarized in Table 2. The full scheme, including self-adaptive loss balancing, achieves the best performance, reaching a relative $L^2$ error of $8.29 \times 10^{-4}$ and relative $L^1$ error of $5.34 \times 10^{-4}$. In particular, using the faster-decaying initial condition ($IC1$) leads to better results than the slower-decaying case ($IC2$), suggesting that the initial condition plays a more critical role during the early stages of training but becomes less dominant over time. The ablation results further highlight the importance of each methodological component. Removing any single part generally leads to performance degradation, with the most severe impact observed when the initial condition is omitted, resulting in a relative error close to 0.99 in both $L^1$ and $L^2$ norms. Figure 4 presents cross-sectional comparisons at different times, further confirming the excellent agreement between the predicted and true solutions across the entire time horizon. These results provide strong evidence for the effectiveness of the proposed training strategy and underscore the critical role of each pipeline component.

| Ablation Settings | | | | Case $I$ | | Case $II$ | |
|---|---|---|---|---|---|---|---|
| PBC | IC1 | IC2 | SA | Rel. $L^2$ error | Rel. $L^1$ error | Rel. $L^2$ error | Rel. $L^1$ error |
| ✓ | ✓ | ✗ | ✓ | $8.29 \times 10^{-4}$ | $5.34 \times 10^{-4}$ | $3.53 \times 10^{-4}$ | $2.14 \times 10^{-4}$ |
| ✓ | ✗ | ✓ | ✓ | $7.64 \times 10^{-2}$ | $1.66 \times 10^{-2}$ | $7.04 \times 10^{-4}$ | $3.26 \times 10^{-4}$ |
| ✓ | ✓ | ✗ | ✗ | $1.24 \times 10^{-3}$ | $6.10 \times 10^{-4}$ | $8.32 \times 10^{-4}$ | $3.72 \times 10^{-4}$ |
| ✓ | ✗ | ✓ | ✗ | $2.86 \times 10^{-2}$ | $8.20 \times 10^{-3}$ | $9.80 \times 10^{-4}$ | $3.91 \times 10^{-4}$ |
| ✓ | ✗ | ✗ | ✗ | $9.99 \times 10^{-1}$ | $9.99 \times 10^{-1}$ | $9.99 \times 10^{-1}$ | $9.99 \times 10^{-1}$ |
| ✓ | ✗ | ✗ | ✓ | $9.99 \times 10^{-1}$ | $9.99 \times 10^{-1}$ | $9.99 \times 10^{-1}$ | $9.99 \times 10^{-1}$ |
| ✗ | ✓ | ✗ | ✗ | $1.05 \times 10^{-2}$ | $2.65 \times 10^{-3}$ | $5.11 \times 10^{-2}$ | $1.21 \times 10^{-2}$ |
| ✗ | ✓ | ✗ | ✓ | $1.86 \times 10^{-3}$ | $7.31 \times 10^{-4}$ | $5.03 \times 10^{-2}$ | $8.27 \times 10^{-3}$ |
| ✗ | ✗ | ✓ | ✗ | $1.88 \times 10^{-2}$ | $4.71 \times 10^{-3}$ | $4.58 \times 10^{-2}$ | $1.10 \times 10^{-2}$ |
| ✗ | ✗ | ✓ | ✓ | $4.34 \times 10^{-3}$ | $1.24 \times 10^{-3}$ | $3.03 \times 10^{-2}$ | $4.25 \times 10^{-3}$ |
| ✗ | ✗ | ✗ | ✓ | $5.12 \times 10^{-1}$ | $3.18 \times 10^{-1}$ | $4.02 \times 10^{-1}$ | $1.42 \times 10^{-1}$ |
| ✗ | ✗ | ✗ | ✗ | $9.99 \times 10^{-1}$ | $9.98 \times 10^{-1}$ | $5.11 \times 10^{-1}$ | $3.28 \times 10^{-1}$ |

Table 2: *Allen-Cahn (Casse I and Case II):* Relative $L^2$ and $L^1$ errors for an ablation study illustrating the impact of disabling individual components of the proposed technique and training pipeline.

### D.2.2 CASE $II$

For the second case, we change the IC to have highly oscillatory data by $u(0, x) = x^2 \sin(2\pi x)$. Other parameters are $T = 1$, $a = -1$, $b = 1$, $\gamma_1 = 0.001$ and $\gamma_2 = 4$.

We first train the baseline PINN model, which produces a large relative $L^2$ error of $0.511$ and relative $L^1$ error of $0.328$, demonstrating its inadequacy in resolving the sharp interface dynamics (Table 2). However, once the initial condition enforcement and periodic boundary neural networks are incorporated, the performance improves drastically, achieving a relative $L^2$ error of $8.32 \times 10^{-4}$ and relative $L^1$ error of $3.72 \times 10^{-4}$. Further gains are realized when using the self-adaptive loss weighting strategy, although the improvement is marginal compared to the dominant contribution from enforcing the initial and boundary conditions. Interestingly, unlike the previous case, there is minimal performance difference between the schemes using $IC1$ and $IC2$, suggesting that the impact of initial condition decay becomes less significant when strong oscillations are already present at the beginning.

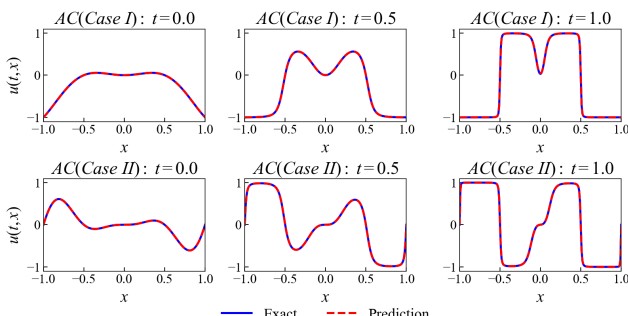

Figure 4: Best solution: *Allen-Cahn*.

### D.3 KURAMOTO-SIVASHINSKY PDE

Kuramoto–Sivashinsky (KS) equation is a fourth-order nonlinear partial differential equation, known for its chaotic behavior Kudryashov (1990); Michelson (1986). It has served as a pivotal model in the study of complex dynamical phenomena observed in various physical systems. Consequently, the KS equation provides an excellent case study to demonstrate how PINNs can effectively simulate chaotic dynamics. The equation is given by:

$$u_t = -u_{xx} - u_{xxxx} - uu_x, \quad (t,x) \in (0,T] \times (a,b),$$

$$u(0,x) = \cos(\frac{x}{16})(1 + \sin\frac{x-1}{16}), \quad x \in [a,b], \quad u(t,a) = u(t,b), \ t \in [0,T], \tag{8}$$

where $T = 20, a = 0, b = 32\pi$. The best results are shown in 5:

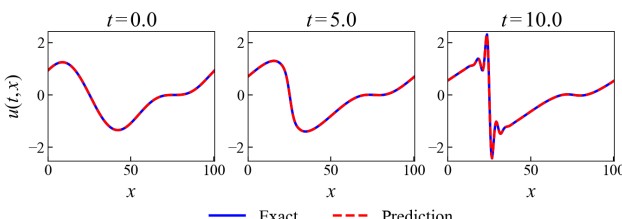

Figure 5: Best solution: *Kuramoto-Sivashinsky*.

As in Table 3, solving this PDE system using a baseline PINN alone proves to be exceptionally challenging. The incorporation of periodic neural networks for boundary conditions, paired with initial condition *IC2*, consistently yields superior results compared to *IC1*. This suggests that, in contrast to the Allen-Cahn equation discussed in Section D.2, the choice of initial condition plays a more significant role in the training process for this more complex system. This observation underscores the importance of carefully selecting both the boundary conditions and initial conditions when training neural networks for chaotic or higher-order PDE systems, where these factors contribute notably to the accuracy and stability of the solution.

### D.4 CAHN-HILLIARD EQUATION

The Cahn-Hilliard equation, a variant of the Allen-Cahn equation, also models phase separation. This process describes how the components of a binary fluid spontaneously separate into distinct domains, each composed of a single component Cahn & Hilliard (1958); Kim et al. (2016); Lee et al. (2014); Miranville (2017). The equation is expressed as:

$$u_t = \epsilon_1(-u_{xx} - \epsilon_2 u_{xxxx} + (u^3)_{xx}), \quad (t,x) \in (0,T] \times (-L,L),$$

$$u(0,x) = u_0(x), \quad x \in [-L,L], \tag{9}$$

$$u(t,-L) = u(t,L), \quad t \in [0,T],$$

| Ablation Settings | | | | Performance | |
|:---:|:---:|:---:|:---:|:---:|:---:|
| **PBC** | **IC1** | **IC2** | **SA** | **Rel. $L^2$ error** | **Rel. $L^1$ error** |
| ✓ | ✓ | ✗ | ✓ | $4.04 \times 10^{-2}$ | $1.46 \times 10^{-2}$ |
| ✓ | ✗ | ✓ | ✓ | $5.37 \times 10^{-4}$ | $1.02 \times 10^{-3}$ |
| ✓ | ✓ | ✗ | ✗ | $1.68 \times 10^{-2}$ | $8.94 \times 10^{-3}$ |
| ✓ | ✗ | ✓ | ✗ | $1.30 \times 10^{-2}$ | $7.79 \times 10^{-3}$ |
| ✓ | ✗ | ✗ | ✗ | $9.84 \times 10^{-3}$ | $6.36 \times 10^{-3}$ |
| ✓ | ✗ | ✗ | ✓ | $2.50 \times 10^{-3}$ | $1.81 \times 10^{-3}$ |
| ✗ | ✓ | ✗ | ✗ | $2.13 \times 10^{-1}$ | $9.74 \times 10^{-2}$ |
| ✗ | ✓ | ✗ | ✓ | $2.00 \times 10^{-1}$ | $1.04 \times 10^{-1}$ |
| ✗ | ✗ | ✓ | ✗ | $1.54 \times 10^{-2}$ | $1.21 \times 10^{-2}$ |
| ✗ | ✗ | ✓ | ✓ | $9.60 \times 10^{-3}$ | $8.75 \times 10^{-3}$ |
| ✗ | ✗ | ✗ | ✓ | $2.11 \times 10^{-1}$ | $1.06 \times 10^{-1}$ |
| ✗ | ✗ | ✗ | ✗ | $3.01 \times 10^{-1}$ | $1.97 \times 10^{-1}$ |

Table 3: *Kuramoto-Sivashinsky:* Relative $L^2$ and $L^1$ errors for an ablation study illustrating the impact of disabling individual components of the proposed technique and training pipeline.

where $\epsilon_1 = 10^{-2}, \epsilon_2 = 10^{-4}, T = 1, L = 1$. We specify the initial condition as $u_0(x) = -\cos(2\pi x)$. This PDE involves higher-order derivatives, making it more challenging to solve compared to the Allen-Cahn equation.

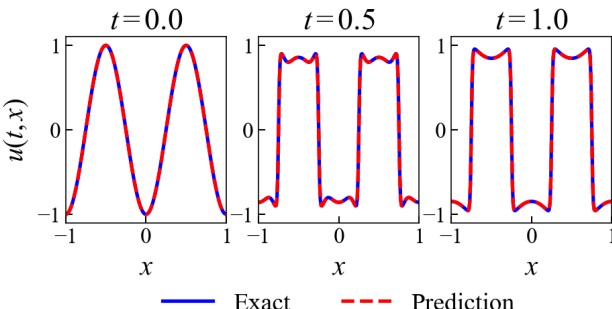

Figure 6: Best solution of the *Cahn-Hilliard*. The resulting relative $L^2$ error is $3.53 \times 10^{-4}$.

In this experiment, the full scheme yields the best results. Additionally, methods using initial condition *IC2* outperform those with *IC1*. Except for the full schemes and the experiment with periodic neural networks for boundary conditions and initial condition *IC2*, none of the other approaches produce accurate solutions. The errors for all experiments are summarized in Table 4. The best results are illustrated in Figure 6.

### D.5 GRAY-SCOTT EQUATION

Reaction and diffusion of chemical species can produce a variety of patterns (De Kepper et al., 1991; Nishiura & Ueyama, 1999), reminiscent of those often seen in nature. The Gray-Scott type system is one of classical mathematical models for chemical reactions (Gray & Scott, 1983; 1984; Liang et al., 2022). The general irreversible Gray-Scott equations describe such reactions:

$$U + 2V \longrightarrow 3V,$$
$$V \longrightarrow P. \tag{10}$$

| Ablation Settings | | | | Performance | |
|:---:|:---:|:---:|:---:|:---:|:---:|
| **PBC** | **IC1** | **IC2** | **SA** | **Rel. $L^2$ error** | **Rel. $L^1$ error** |
| ✓ | ✓ | ✗ | ✓ | $1.06 \times 10^{-3}$ | $5.61 \times 10^{-4}$ |
| ✓ | ✗ | ✓ | ✓ | $9.75 \times 10^{-4}$ | $5.65 \times 10^{-4}$ |
| ✓ | ✓ | ✗ | ✗ | $2.50 \times 10^{-1}$ | $5.87 \times 10^{-2}$ |
| ✓ | ✗ | ✓ | ✗ | $3.87 \times 10^{-3}$ | $1.91 \times 10^{-3}$ |
| ✓ | ✗ | ✗ | ✗ | $9.99 \times 10^{-1}$ | $9.99 \times 10^{-1}$ |
| ✓ | ✗ | ✗ | ✓ | $9.99 \times 10^{-1}$ | $9.99 \times 10^{-1}$ |
| ✗ | ✓ | ✗ | ✗ | $3.48 \times 10^{-1}$ | $2.94 \times 10^{-1}$ |
| ✗ | ✓ | ✗ | ✓ | $2.37 \times 10^{-1}$ | $1.09 \times 10^{-1}$ |
| ✗ | ✗ | ✓ | ✗ | $4.07 \times 10^{-1}$ | $3.45 \times 10^{-1}$ |
| ✗ | ✗ | ✓ | ✓ | $1.55 \times 10^{-2}$ | $8.20 \times 10^{-3}$ |
| ✗ | ✗ | ✗ | ✓ | $9.99 \times 10^{-1}$ | $9.99 \times 10^{-1}$ |
| ✗ | ✗ | ✗ | ✗ | $9.99 \times 10^{-1}$ | $9.98 \times 10^{-1}$ |

Table 4: *Cahn-Hilliard Equation:* Relative $L^2$ and $L^1$ errors for an ablation study illustrating the impact of disabling individual components of the proposed technique and training pipeline.

This system is defined by two equations that describe the dynamics of two reacting substances:

$$
\begin{aligned}
u_t &= \epsilon_1 u_{xx} + b(1-u) - uv^2, \\
v_t &= \epsilon_2 v_{xx} - (b+k)v + uv^2, \quad (t,x) \in (0,T] \times (-L,L), \\
u(0,x) &= u_0(x), \quad v(0,x) = v_0(x), \quad x \in [-L,L], \\
u(t,-L) &= u(t,L), \quad v(t,-L) = v(t,L), \quad t \in [0,T],
\end{aligned}
\tag{11}
$$

where $T = 20, L = 50, \epsilon_1 = 1, \epsilon_2 = 0.01$ are diffusion rates, $b = 0.02$ is the "feeding rate" that adds $U$, $k = 0.0562$ is the "killing rate" that removes $V$. We set our initial conditions as:

$$
u_0(x) = 1 - \frac{\sin(\pi(x - 50)/100)^4}{2},
$$

$$
v_0(x) = \frac{\sin(\pi(x - 50)/100)^4}{4}.
$$

In this ablation study, we not only evaluate the performance of self-adaptive neural networks but also test our scheme with mini-batching on the Gray-Scott equations, which model the densities of two interacting species. For mini-batching, we use 5 batches, each containing 3,280 samples. As shown in Table 5, the results confirm that each proposed component improves the overall model performance. Omitting any one of these components leads to higher error rates. Notably, the errors for the $u-$equation are typically smaller than those for the $v-$equation, which aligns with our expectations. This is because the $v-$equation is sharper and more challenging to train compared to the $u-$equation. Additionally, experiments using initial condition *IC2* consistently outperform those using *IC1*. Specifically, the full scheme with *IC2* achieves the best results, with a relative $L^2$ error of $2.05 \times 10^{-4}$ for $u-$equation and $8.46 \times 10^{-4}$ for $v-$equation. These results suggest that *IC2* is more effective for larger time-scale PDEs, as *IC1* causes the initial condition to decay too quickly, reducing its influence during training. The predicted solutions from our best model are visualized in Figures 7 and 8, respectively.

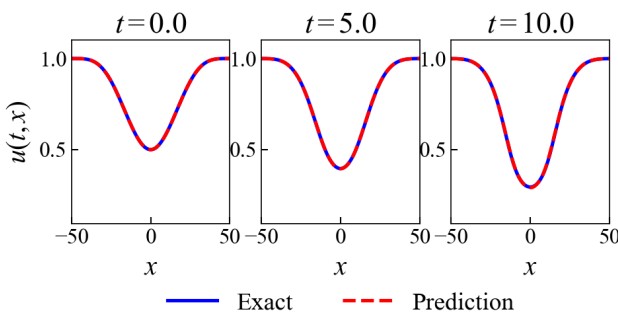

Figure 7: Best solution of the *Gray-Scott (u species)*. The resulting relative $L^2$ error is $2.05 \times 10^{-4}$.

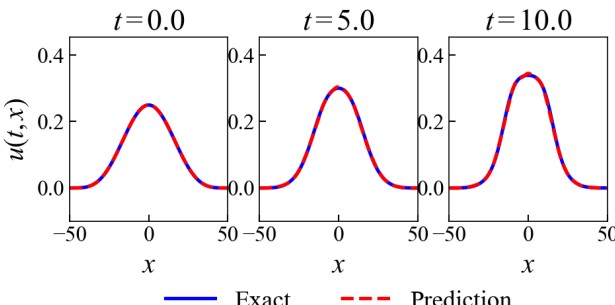

Figure 8: Best solution of the *Gray-Scott (v species)*. The resulting relative $L^2$ error is $8.46 \times 10^{-4}$.

| Ablation Settings | | | | | Performance | |
|---|---|---|---|---|---|---|
| **PBC** | **IC1** | **IC2** | **SA** | **mini-batch** | $\mathcal{E}_2 : u$ | $\mathcal{E}_2 : v$ |
| ✓ | ✓ | ✗ | ✓ | ✓ | $8.98 \times 10^{-4}$ | $7.92 \times 10^{-3}$ |
| ✓ | ✗ | ✓ | ✓ | ✓ | $2.05 \times 10^{-4}$ | $8.46 \times 10^{-4}$ |
| ✓ | ✓ | ✗ | ✗ | ✓ | $4.93 \times 10^{-3}$ | $3.03 \times 10^{-2}$ |
| ✓ | ✗ | ✓ | ✗ | ✓ | $2.86 \times 10^{-3}$ | $8.20 \times 10^{-3}$ |
| ✓ | ✓ | ✗ | ✓ | ✗ | $9.01 \times 10^{-3}$ | $4.27 \times 10^{-3}$ |
| ✓ | ✗ | ✓ | ✓ | ✗ | $7.59 \times 10^{-3}$ | $3.98 \times 10^{-3}$ |
| ✓ | ✓ | ✗ | ✗ | ✗ | $5.11 \times 10^{-3}$ | $2.58 \times 10^{-2}$ |
| ✓ | ✗ | ✓ | ✗ | ✗ | $1.23 \times 10^{-3}$ | $9.74 \times 10^{-3}$ |
| ✓ | ✗ | ✗ | ✗ | ✗ | $8.09 \times 10^{-1}$ | $9.99 \times 10^{-1}$ |
| ✗ | ✓ | ✗ | ✗ | ✗ | $2.06 \times 10^{-1}$ | $7.43 \times 10^{-1}$ |
| ✗ | ✗ | ✓ | ✗ | ✗ | $5.37 \times 10^{-1}$ | $2.58 \times 10^{-1}$ |
| ✗ | ✗ | ✗ | ✓ | ✗ | $1.15 \times 10^{-1}$ | $2.62 \times 10^{-1}$ |
| ✗ | ✗ | ✗ | ✗ | ✓ | $9.53 \times 10^{-2}$ | $8.65 \times 10^{-2}$ |
| ✗ | ✗ | ✗ | ✗ | ✗ | $9.21 \times 10^{-2}$ | $2.98 \times 10^{-1}$ |

Table 5: *Gray-Scott Equation:* Relative $L^2$ errors of $u$ and $v$ equations for an ablation study illustrating the impact of disabling individual components of the proposed technique and training pipeline.

## D.6 NONLINEAR SCHRÖEDINGER EQUATION

In theoretical physics, nonlinear Schröedinger equation is a nonlinear PDE, applicable to classical and quantum mechanics (Kato, 1987; Kevrekidis et al., 2001; Fibich, 2015). The dimensionless equation of the classical field is

$$
\begin{aligned}
u_t &= iu_{xx} + i|u|^2 u, \quad (t,x) \in [0,2] \times [-\pi, \pi], \\
u(0,x) &= u_0(x), \quad x \in [-\pi, \pi], \\
u(t,-\pi) &= u(t,\pi), \quad t \in [0,2],
\end{aligned}
\tag{12}
$$

where we let

$$
u_0(x) = \frac{2}{2 - \sqrt{2}\cos(x)} - 1.
$$

Using the same parameter settings and training steps with prior experiments, we obtain the following results:

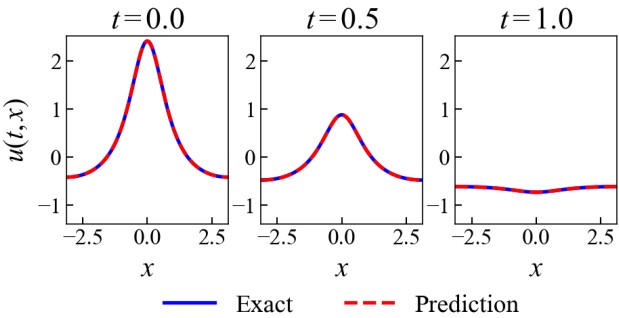

Figure 9: Best real-part solutions of the *Nonlinear Schröedinger Equation*.

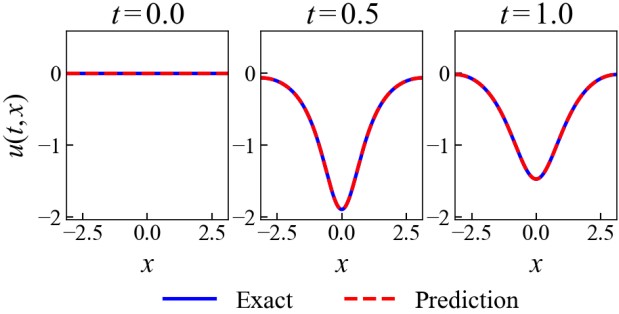

Figure 10: Best imaginary-part solutions of the *Nonlinear Schröedinger Equation*.

Table 6 presents the results of all experiments. The full scheme achieves the best performance, with a relative $L^2$ error of real part of $1.80 \times 10^{-4}$ and the relative $L^2$ error of imaginary part of $2.58 \times 10^{-4}$. There is no significant difference in the errors between methods using *IC1* and those using *IC2*. Moreover, without the use of self-adaptive neural networks or mini-batching, accurately obtaining the solution becomes challenging, highlighting the importance of both methods in ensuring more stable training.

## D.7 LINEAR ADVECTION

The linear advection equation models the transport of a quantity by a constant velocity field and serves as a fundamental benchmark for testing numerical methods due to its simplicity and rich

| Ablation Settings | | | | | Performance: $\mathcal{E}_2$ | |
|:---:|:---:|:---:|:---:|:---:|:---:|:---:|
| PBC | IC1 | IC2 | SA | mini-batch | Real part | Imag part |
| ✓ | ✓ | ✗ | ✓ | ✓ | $1.80 \times 10^{-4}$ | $2.58 \times 10^{-4}$ |
| ✓ | ✗ | ✓ | ✓ | ✓ | $1.84 \times 10^{-4}$ | $3.27 \times 10^{-4}$ |
| ✓ | ✓ | ✗ | ✗ | ✓ | $7.34 \times 10^{-3}$ | $1.23 \times 10^{-2}$ |
| ✓ | ✗ | ✓ | ✗ | ✓ | $8.32 \times 10^{-3}$ | $1.39 \times 10^{-2}$ |
| ✓ | ✓ | ✗ | ✓ | ✗ | $1.10 \times 10^{-3}$ | $2.10 \times 10^{-3}$ |
| ✓ | ✗ | ✓ | ✓ | ✗ | $1.70 \times 10^{-3}$ | $3.27 \times 10^{-3}$ |
| ✓ | ✓ | ✗ | ✗ | ✗ | $1.00 \times 10^{-1}$ | $1.83 \times 10^{-1}$ |
| ✓ | ✗ | ✓ | ✗ | ✗ | $9.56 \times 10^{-2}$ | $1.56 \times 10^{-1}$ |
| ✓ | ✗ | ✗ | ✗ | ✗ | $8.09 \times 10^{-1}$ | $9.99 \times 10^{-1}$ |
| ✗ | ✓ | ✗ | ✗ | ✗ | $9.99 \times 10^{-1}$ | $8.78 \times 10^{-1}$ |
| ✗ | ✗ | ✓ | ✗ | ✗ | $5.04 \times 10^{-2}$ | $1.21 \times 10^{-1}$ |
| ✗ | ✗ | ✗ | ✓ | ✗ | $5.44 \times 10^{-2}$ | $7.26 \times 10^{-2}$ |
| ✗ | ✗ | ✗ | ✗ | ✓ | $6.60 \times 10^{-2}$ | $1.05 \times 10^{-1}$ |
| ✗ | ✗ | ✗ | ✗ | ✗ | $3.63 \times 10^{-1}$ | $5.06 \times 10^{-1}$ |

Table 6: *Nonlinear Schröedinger Equation:* Relative $L^2$ errors of real and imaginary equations for an ablation study illustrating the impact of disabling individual components of the proposed technique and training pipeline.

structure LeVeque (2002); Toro (2013). It describes the propagation of a scalar profile without deformation and is given by:

$$
\begin{aligned}
u_t + c u_x &= 0, && (t, x) \in (0, T] \times (0, 2\pi), \\
u(0, x) &= \sin(x), && x \in [0, 2\pi], \\
u(t, 0) &= u(t, 2\pi), && t \in [0, T].
\end{aligned}
\tag{13}
$$

Unlike dissipative PDEs such as the Allen–Cahn or Cahn–Hilliard equations, the linear advection equation conserves the shape of the initial profile as it translates. This problem becomes "stiffer" as the advection velocity increases.

**Theorem 3.** *Let $u(t, x)$ denote the solution to the linear advection equation (13). With the transformed neural approximation in (4) and $\psi$ and $\eta$ satisfying the conditions in (5), then $u(t, x)$ satisfies the PDE iff $\eta$ satisfies the forced linear transport equation:*

$$
\eta_t + c \eta_x = -(\psi_t + c \psi_x).
$$

*This PDE is well-posed and stable for PINN training provided the following conditions hold:*

- *The function $f(t, x) := -(\psi_t + c \psi_x)$ satisfies $f \in C^1([0, T] \times \Omega)$.*

- *If $f(t, x)$ is uniformly bounded or decays in time, then the solution $\eta$ remains stable and avoids growth-induced overfitting.*

For our choice, we have $\psi(t, x) = u_0(x) e^{-\alpha t}$ and $\phi(t, x) = t$ with $\alpha > 0$, ensuring $\eta(0, x) = 0$. Then, the forcing term satisfies $\eta_t + c \eta_x = (\alpha \sin x - c \cos x) e^{-\alpha t}$. Since this forcing is smooth and decays exponentially in time, it guarantees both the stability and regularity of the neural network $\eta(t, x)$. Here, we compare the results produced by our method with those from a baseline PINN under the same hyperparameter settings. In our experiments, both models were able to converge at low velocities; however, we observed that higher advection velocities introduced stiffness. Our method exhibited greater robustness and was less affected by stiffness compared to the baseline PINN.

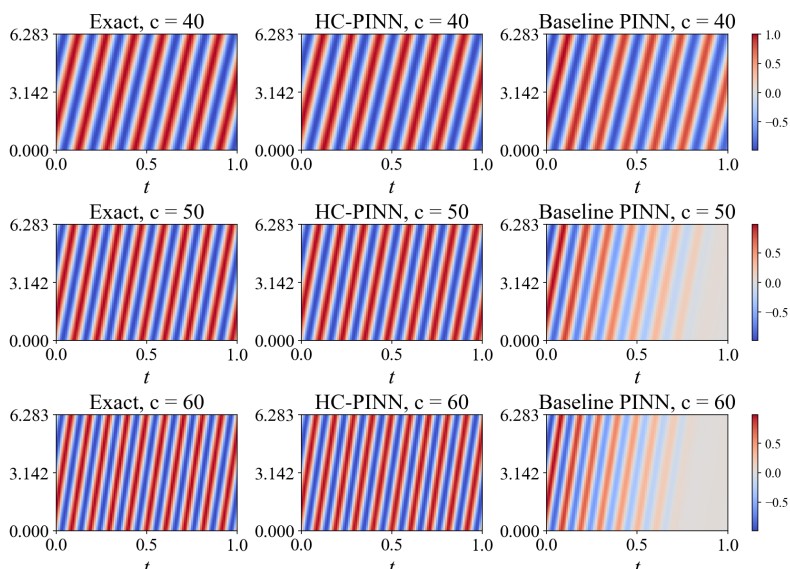

Figure 11: Periodic advection results after 50,000 iterations.

The implementation details are the same as in Section D.1, except that we perform 10 independent replications of the experiment to account for the randomness in neural network weight initialization, rather than using the same initial weights. Table 7 shows the averaged relative $L_1/L_2$-error over 10 independent repetitions. As the advection velocity increases and the PDE becomes stiffer, making it harder for baseline PINN to train. However, our method consistently produces better results. This trend is clearly reflected in Table 7. Even at low velocities, our method demonstrates significantly better accuracy compared to the baseline PINN. As $c$ increases, both the relative $L_1$ and $L_2$ errors grow, however, the errors associated with the PINN rise significantly faster than those of our method. The most striking example occurs at $c = 50$, where the base line PINN fails to approximate the solution, while our method still produces stable and good results. This behavior is evident in Table 7 and Figure 11. Table 8 shows the ablation study of linear advection equation.

| Solver | $c$ | Rel. $L_1$ error | Rel. $L_2$ error |
|---|---|---|---|
| HC-PINN | 30 | 0.0030 | 0.0041 |
| | 40 | 0.0141 | 0.0159 |
| | 50 | 0.0147 | 0.0164 |
| | 60 | 0.0171 | 0.0182 |
| CINN(Braga-Neto, 2022) | 30 | 0.0478 | 0.0579 |
| | 40 | 0.0714 | 0.0852 |
| | 50 | 0.4692 | 0.5365 |
| | 60 | 0.7293 | 0.7134 |
| Baseline PINN | 30 | 0.0873 | 0.1003 |
| | 40 | 0.3918 | 0.4395 |
| | 50 | 0.7140 | 0.7797 |
| | 60 | 0.8339 | 0.8664 |

Table 7: Comparison of HC-PINN, CINN and baseline PINN across different advection speeds $c$ over 10 independent repetitions.

| Ablation Settings | | | | Performance |
| --- | --- | --- | --- | --- |
| **PBC** | **IC** | **SA** | **Rel. $L^2$ error** | **Rel. $L^1$ error** |
| ✓ | ✓ | ✓ | $1.82 \times 10^{-2}$ | $1.03 \times 10^{-2}$ |
| ✓ | ✓ | ✗ | $9.75 \times 10^{-2}$ | $8.65 \times 10^{-2}$ |
| ✓ | ✗ | ✓ | $7.50 \times 10^{-1}$ | $7.87 \times 10^{-1}$ |
| ✗ | ✓ | ✓ | $3.87 \times 10^{-2}$ | $5.47 \times 10^{-2}$ |
| ✓ | ✗ | ✗ | $7.44 \times 10^{-1}$ | $8.01 \times 10^{-1}$ |
| ✗ | ✓ | ✗ | $1.07 \times 10^{-1}$ | $1.19 \times 10^{-1}$ |
| ✗ | ✗ | ✓ | $7.65 \times 10^{-1}$ | $7.71 \times 10^{-1}$ |
| ✗ | ✗ | ✗ | $8.66 \times 10^{-1}$ | $8.34 \times 10^{-1}$ |

Table 8: *Linear Advection Equation (c=60):* Relative $L^2$ and $L^1$ errors for an ablation study illustrating the impact of disabling individual components of the proposed technique and training pipeline.

