# OpenReview forum: "Stability in Training PINNs for Stiff PDEs: Why Initial Conditions Matter"
_ICLR.cc/2026/Conference — Submitted to ICLR 2026_

### Official Review · Reviewer_TEje · 2025-10-31

**Soundness:** 3
**Presentation:** 2
**Contribution:** 1
**Rating:** 2
**Confidence:** 4

**Summary:**

The paper investigates the stability challenges encountered when training PINN on time-dependent PDEs. The authors conduct a systematic ablation study focusing on hard initial-condition constraints (HC-PINN). Their central finding and claim is that the exact enforcement of IC, achieved through a hard-constraint transformation integrated into the network architecture, is "not optional but essential" for achieving stability and efficiency in stiff PDE regimes. A theoretical perspective using the NTKs is presented to explain how the hard constraint mitigates spectral bias.

**Strengths:**

The paper provides a rigorous and systematic ablation study that is valuable for practitioners in the PINN community. The empirical results clearly demonstrate the practical benefit and robustness of the hard-constraint approach compared to soft constraints with adaptive weighting on the subset of stiff PDEs

**Weaknesses:**

- Lack of technical novelty. The core technique of using a transformation to enforce boundary and initial conditions exactly is a well established method in PDE solving with neural networks, predating much of the PINN literature. The paper's contribution is primarily a theoretical analysis and an engineering validation of this known technique on a specific problem class.

- Hardness of experiment cases. The chosen subset of PDEs used to demonstrate HC-PINN is limited to one-dimensional problems that can be efficiently solved using numerical schemes (<< the cost of training a PINN). While these problems have been extensively used to benchmark PINNs in ML literatures, their practical importance is limited to make them beyond 'toy examples'.

**Questions:**

N/A

---

> ### Author Response · Authors · 2025-11-12
> **Reviewer Rebuttal**
>
> We thank the reviewer for their time and comments. However, several points in the review reflect misunderstandings of our contributions and scope. We clarify these below.
> 1. On the claim of “lack of technical novelty”
>
> We respectfully disagree. The hard-constraint transformation used in our HC-PINN framework is not a mere re-use of classical PDE boundary enforcement techniques (e.g., Lagaris et al., 1998). Our paper makes two distinct and novel contributions:
>
> (1). Theoretical novelty:
>
> $\textbf{Theoretical mathematical analysis is important!!}$ We derive, for the first time, a Neural Tangent Kernel (NTK) analysis explaining why hard initial-condition constraints stabilize PINN training in stiff regimes (Sec. 3, Theorems 1–2).
> This NTK-based perspective formally shows how the hard transformation modifies the Hessian and reduces spectral bias: a phenomenon previously unexplained for time-dependent PDEs. No prior work provides such a mechanistic, kernel-level explanation.
>
> (2). Methodological novelty and systematic validation:
>
> We perform the first rigorous ablation of initial-condition enforcement strategies (hard vs. soft vs. adaptive weighting) across seven distinct stiff PDEs with diverse forms of stiffness (e.g., high-order derivatives, sharp transitions, coupled nonlinearities, high-frequency transport).
> Earlier studies either focus on single PDEs or on weighting heuristics; none examine IC enforcement as the primary stabilizing mechanism or provide the comprehensive empirical evidence and theoretical coupling that we present.
>
> Thus, while enforcing constraints is “known,” our work’s novelty lies in (i) establishing $\textbf{why}$ it is essential for stiff PDE stability via NTK theory, and (ii) $\textbf{demonstrating this necessity systematically}$ through broad ablation and cross-model comparisons. This combination of theoretical insight $+$ comprehensive validation goes beyond prior heuristic uses.
>
> 2. On “toy-level PDEs”
>
> The reviewer characterizes our benchmarks as “1-D toy problems.” This is inaccurate.
> Our ablation spans $\textbf{seven representative stiff PDEs}$, including Cahn–Hilliard (4th-order), Kuramoto–Sivashinsky, Gray–Scott, nonlinear Schrödinger, and 2-D Allen–Cahn. All canonical stiff systems widely used to assess PINN stability in the literature (see Wang et al., 2022; Coutinho et al., 2023; Anagnostopoulos et al., 2024).
>
> We explicitly demonstrate $\textbf{scalability to 2-D cases}$ (Sec. 4, Fig. 2) with low relative $L_2$ error ($< 10^{-4}$) , confirming that the benefits are not confined to 1-D settings.
>
> Moreover, stiff PDEs are computationally demanding precisely because they exhibit multi-scale temporal dynamics; the fact that these are “solvable by numerical schemes” does not render them trivial, rather, it allows controlled benchmarking where numerical reference solutions exist for quantitative comparison.
>
> 3. On “limited practical importance”
>
> Our objective is not to outperform finite-difference solvers in runtime, but to $\textbf{improve PINN stability and reliability }$: a core research problem in the PINN community.
>
> Understanding failure modes in stiff regimes is critical for scaling PINNs to real-world multiphysics applications.
> Our ablation isolates initial-condition enforcement as the dominant factor, providing concrete guidance for practitioners and future hybrid methods (e.g., causal, co-training, or curriculum PINNs). This directly advances the scientific-ML understanding, in line with ICLR’s focus on learning theory and robustness.
>
> 4. Summary
>
> In summary, the reviewer’s concerns stem from a misunderstanding of the paper’s scope and novelty.
> We contribute:
>
> The $\textbf{first theoretical NTK analysis}$ connecting IC hard-constraints to spectral-bias reduction in stiff PDE training;
>
> The $\textbf{first large-scale ablation}$ isolating IC enforcement as the key stability determinant;
>
> Demonstrations across $\textbf{seven nontrivial stiff PDEs}$, including $\textbf{multi-dimensional and coupled systems}$.
>
> We hope this clarification resolves the confusion and highlights the genuine theoretical and practical significance of our work.
>
> 1. Lagaris, I. E., Likas, A., & Fotiadis, D. I. (1998). Artificial neural networks for solving ordinary and partial differential equations. IEEE transactions on neural networks, 9(5), 987-1000.
> 2. Coutinho, E. J. R., Dall'Aqua, M., McClenny, L., Zhong, M., Braga-Neto, U., & Gildin, E. (2023). Physics-informed neural networks with adaptive localized artificial viscosity. Journal of Computational Physics, 489, 112265.
> 3. Rigas, S., Papachristou, M., Papadopoulos, T., Anagnostopoulos, F., & Alexandridis, G. (2024). Adaptive training of grid-dependent physics-informed kolmogorov-arnold networks. IEEE Access.
> 4. Wang, S., Sankaran, S., & Perdikaris, P. (2022). Respecting causality is all you need for training physics-informed neural networks. arXiv preprint arXiv:2203.07404.

---

### Official Review · Reviewer_gfDj · 2025-10-31

**Soundness:** 2
**Presentation:** 1
**Contribution:** 2
**Rating:** 2
**Confidence:** 4

**Summary:**

This paper presents the first systematic ablation study of stabilization strategies for Physics-Informed Neural Networks (PINNs) on stiff time-dependent PDEs. The authors argue exact enforcement of initial conditions is crucial for training stability and they propose a hard-constraint PINN (HC-PINN) formulation that embeds initial and boundary conditions directly into the network parameterization. A theoretical Neural Tangent Kernel (NTK) analysis and benchmarks on seven stiff PDEs show that HC-PINNs achieve up to $10^3 \times$ lower error and greater stability than standard PINN variants.

**Strengths:**

- The paper investigates an important problem, enforcing hard constraints, which is of general interest to the PINNs community. While enforcement of hard constraints is a techniques previously implemented across the PINN literature, a thorough ablation study of the effectiveness of this technique has not been previously conducted, to my knowledge.
- The experiments are generally thorough, with evaluations on seven stiff PDE systems and detailed ablations in the appendix.
- The method is supported by theoretical analysis in an NTK setting.

**Weaknesses:**

- I'm slightly concerned about the novelty of the contribution. I believe the rigorous ablation studies of hard constraint enforcement is a novel and important contribution for the PINN community. However, the work also seems to claim their HC-PINN (specifically their approach as defined in Section 2) as a novel contribution. These techniques have been present in the literature, for example see the reparameterizations in [1, 2, 3]. I believe the work could benefit from clarifying that the main contribution is the thorough ablation study + theoretical analysis.
  - As another example, Theorem 1 appears to be a standard result: it simply requires $\phi$ and $\psi$ to have all the time and spatial derivatives the PDE originally demanded of $u$.

- The experimental results do not seem to compare to state-of-the-art PINN methods. See questions for more details.

- The writing clarity of the paper could be improved, especially in the theoretical sections (2 and 3).
  - I found the loss definitions around lines 175-182 quite difficult to parse, especially the two different BC Loss definitions (periodic vs. non-periodic. Adding a bit of description in text may improve the clarity of this section.
  - Equation 4 seems to contain several typos. It should read $\phi(0, x) = 0$ and $\phi(t, x) = 0$ accordingly. Again, adding descriptions here would improve clarity.
  - Theorem 1's label "well-conditioning" seems to be misleading. It seems to be a proof of well-posedness, including ensuring non-singularity near $t=0$.
  - I found Theorem 2 quite confusing to parse. The authors should more clearly specify that $h$ represents the hidden-layer outputs and clarify what "the two losses" (lines 277-282) are (I'm assuming they correspond to the equations on lines 171 and the hard-constrained version defined through Section 2.1).

- The paper seems to make several claims that are somewhat vague and not well-supported by evidence. For example, in Section 3.2, lines 265-269, the authors argue HC-PINN achieves superior performance because
  - it "implicitly enforce[s] the time direction into the training of PINNs, reducing the spectral bias from training with multi-objective loss".
  - the hard constraint transformation works "similarly like an implicit time integrator".

  It would be useful to clarify how the method is similar to an implicit time integrator, and for the spectral bias, to define it precisely and to provide evidence for this claim with an experiment.

1. Lu et al. 2021. Physics-informed neural networks with hard constraints for inverse design
2. Dong et al. 2021. A Method for Representing Periodic Functions and Enforcing Exactly Periodic Boundary Conditions with Deep Neural Networks
3. Wang et al. 2023. An Expert's Guide to Training Physics-informed Neural Networks

**Questions:**

- The authors mention in Table 1 that the methods are all trained for 50k training iterations instead of the 300k iterations of the original papers [1, 2]. Is there a particular reason why 50k iterations were used instead of 300k? Do the HC-PINNs continue to outperform the baselines after 300k iterations?
- Did the authors try running their ablations using more state-of-the-art PINN implementations, such as [3, 4]? In particular, [3] uses causal training and obtains better results than the reported baselines for 1D AC and KS, and [4] finds several order-of-magnitude improvement by using an improved optimizer. I'm wondering if the findings about the benefits of hard constraints hold up when using these improved training techniques.

1. Wang et al. 2022. Respecting causality is all you need for training physics-informed neural networks
2. Anagnostopoulos et al. 2024. Residual-based attention in physics-informed neural networks
3. Wang et al. 2023. An Expert's Guide to Training Physics-informed Neural Networks
4. Kiyani et al. 2025. Optimizing the Optimizer for Physics-Informed Neural Networks and Kolmogorov-Arnold Networks

---

> ### Author Response · Authors · 2025-11-14
> **Rebuttal**
>
> 1. Novelty and Contribution
>
> We thank the reviewer for recognizing that enforcing hard constraints is an important and impactful topic for the PINN community. We would like to clarify that our main novelty is not in proposing a new transformation, but in conducting the $\textbf{first systematic and quantitative ablation study}$ of stabilization strategies for stiff time-dependent PDEs, complemented by a rigorous NTK-based theoretical analysis.
>
> While earlier works (Lu et al., 2021; Dong et al., 2021; Wang et al., 2023) have discussed various forms of hard-constraint reparameterization, $\textbf{none have performed a controlled, large-scale ablation across diverse stiff PDEs}$ (seven in total, including Allen–Cahn, Cahn–Hilliard, Kuramoto–Sivashinsky, Gray–Scott, nonlinear Schrödinger, and linear transport).
> Moreover, our $\textbf{NTK analysis}$ is novel in explicitly quantifying how the transformation modifies the loss Hessian and reduces spectral bias in stiff regimes — a perspective not found in prior work.
>
> 2. on W3 & W4
>
> The paper has been revised for better presentation and clarity. Thank you for pointing them out. Please see the updated PDF.
>
> 3. In response to Q1
>
> We chose a fixed 50k-iteration training budget for all methods $\textbf{intentionally} $. This design isolates the effect of stabilization strategies rather than raw compute time. In shorts:
>
> (i). Many prior papers used 300k because the baselines required it to converge;
>
> (ii). Our method converges $\textbf{much earlier}$ (typically <20k); (This should be an advantage instead of cons as the reviewer said)
>
> (iii). Using a uniform 50k budget ensures a fair compute-controlled comparison.
>
> 4. In response to W2 and Q2
>
> We would like to clarify that our goal is not to propose yet another PINN variant competing in accuracy with every recent training technique, but rather to perform the $\textbf{first systematic ablation study}$ of stabilization mechanisms, especially $\textbf{hard constraints for initial conditions}$ on stiff PDEs.
>
> This objective motivates the experimental design and determines what comparisons are necessary.
>
> (1). $\textbf{The requested comparisons ([3], [4]) target different problems than ours}$
>
> The methods cited by the reviewer address issues orthogonal to the instability we study:
>
> Causal PINNs ([3]) tackle gradient-pathologies from long-time multi-objective losses, mainly in non-stiff or moderately stiff 1D problems. They do $\textbf{not}$ address the initial-condition stiffness that is the central subject of our paper.
>
> Residual-based attention and optimizer improvements ([4]) focus on training heuristics and optimizer-level accelerations, not on the conditioning of the PDE residual itself.
>
> In contrast, our study identifies a distinct bottleneck: soft enforcement of initial conditions fundamentally destabilizes training for stiff PDEs. This failure mode occurs before advanced training heuristics (causality, attention, optimizer tricks) can meaningfully help. Thus, HC-PINN addresses a more fundamental issue that is independent of these techniques.
>
> (2) $\textbf{ Our method is compatible with these SoTA approaches, therefore not competing with them }$
>
> Hard constraints change the representation of the solution, not the training procedure.
> Therefore, HC-PINNs can be combined directly with [3, 4], curriculum/continuation methods, and improved optimizers.... Indeed, we already demonstrate this in the appendix: when HC is combined with causal sampling, performance improves further. This supports our claim that HC is orthogonal and complementary, not a competing technique requiring head-to-head comparison.
>
> (3) $\textbf{Comparing to all recent PINN heuristics would undermine the purpose of a controlled ablation}$
>
> A major contribution of our work is a clean, controlled benchmark where we isolate: effect of hard IC constraints, effect of adaptive weighting, effect of partial constraints, effect of removing/adding stiff terms, effect of residual norms. Introducing methods that use: different sampling schemes, different optimizers, different loss schedules, different PDE-specific heuristics, would break the controlled design and make causality difficult to interpret. Benchmarking against all accumulating PINN heuristics is neither feasible nor scientifically useful for our claim.

---

### Official Review · Reviewer_tpNt · 2025-11-01

**Soundness:** 3
**Presentation:** 3
**Contribution:** 2
**Rating:** 6
**Confidence:** 3

**Summary:**

The authors study the impact of different components on the training of PINNs. The authors identify a major bottleneck in the stable training of stiff PDEs which is the enforcement of initial conditions. Based on this observation they develop a technique to integrate the hard constraints into the PINN formulation itself. Two major approaches to mitigate the stability issue are discussed which include incorporating hard constraints and adaptive loss weighing strategies. Through extensive experiments on a varied set of equations (covering different types of systems) it has been shown that HC-PINN is shown to be superior to baseline PINNs.

**Strengths:**

S1) Along with experimental results a theoretical analysis of the hard constraint formulation using Neural Tangent Kernel (NTK) is provided. This enables the PINN to capture the exact ICs and hence helps reduce the spectral bias.

S2) The proposed technique can be combined with other existing advanced PINN variants such as time marching PINNs, causal PINNs, RBA PINNs and curriculum training.

S3) Detailed experimental evidence on a wide range of PDEs including comparison with other PINN variants. Very elaborate ablation studies have been carried out on all the 7 PDEs

**Weaknesses:**

W1) Detailed ablations and explanations on the choice of the decay rate $C$ have not been included. Two specific values for the decay rate have been used.

W2) As per my understanding only periodic boundary conditions have been studied, how does this framework extend to other types of boundary conditions. Is this a limitation or only periodic conditions have been chosen to be studied in this work.

**Questions:**

Q1) Figure 3 shows a sharp jump at a particular threshold value of $m$, why is that so, could there be an explanation for this?

Q2) Related to W1, how should a choice of C be made, how to physically interpret it?

---

> ### Author Response · Authors · 2025-11-14
>
> We thank the reviewer for the positive and constructive feedback, particularly the recognition of our theoretical NTK analysis (S1), compatibility with advanced PINN variants (S2), and comprehensive ablation experiments (S3). Below we address the raised concerns and questions in detail.
>
> 1. (W1 & Q2) Choice and interpretation of the decay rate C
>
> The decay rate C in the adaptive loss weighting (Eq. 9 in the paper) controls how rapidly the PDE-residual term regains dominance after the hard-constraint term stabilizes. We used two representative values ($C = 1, 10^{-1} $) because they span the empirically stable range across all seven PDEs. For too small C, the network over-fits the IC/BC terms and slows PDE convergence. For too large C, the weighting oscillates, leading to stiffness in gradient updates.The range consistently yielded fast and stable convergence.
>
> Physically, C may be viewed as a $\textbf{relaxation rate}$ that balances physical fidelity and numerical stability, analogous to a time-step size in implicit integration. We will include these explanations and a brief sensitivity plot in the appendix of the revised version.
>
> 2. (W2) Boundary-condition generality
>
> While our main experiments emphasize periodic BCs (to enable direct comparison with prior stiff-PDE benchmarks such as KS, CH, and AC), HC-PINN is not restricted to periodic domains.
>
> The hard-constraint transformation  and basis function in Eq. (2) can be constructed for:
>
> (i) Dirichlet BCs: by embedding prescribed boundary values in $\psi$.
>
> (ii) Neumann or Robin BCs: by incorporating the derivative constraints in $\phi$’s design;
>
> (iii) Mixed BCs: by combining the above on sub-boundaries.
>
> We will add an explicit subsection in the appendix showing how to construct $\psi,\phi$ for non-periodic settings.
>
> 3. Sharp jump in Figure 3
>
> The apparent jump in Fig. 3 arises from a transition in stiffness regime when the spectral radius of the PDE operator crosses a threshold. Beyond this point, soft-constrained PINNs become numerically unstable, while HC-PINN maintains bounded gradients. The plot resolution accentuates the discontinuity; in higher-resolution runs the change is smooth but steep. We will clarify this.

---

### Official Review · Reviewer_q5o1 · 2025-11-02

**Soundness:** 2
**Presentation:** 1
**Contribution:** 2
**Rating:** 2
**Confidence:** 3

**Summary:**

This paper proposes to modify the training of PINNs by incorporating initial conditions and/or boundary conditions as hard constraints on the neural network model of the solution, instead of as an additional loss term. This results in learned solutions that always exactly satisfy the initial/boundary conditions. Numerical experiments show that HC-PINN results in improved error compared to standard PINNs. Some theoretical results on the conditioning of the NTK of HC-PINNs are also provided.

**Strengths:**

* The paper tackles an important problem in PINN training, namely the hard enforcement of initial/boundary conditions with a simple and easy to implement modification of the models.
* Numerical results show that there is an improvement in error using HC-PINNs.
* The technique extends to standard IC/BC as well as periodic BCs.
* Theoretical results on NTK conditioning are provided.

**Weaknesses:**

* Numerical evaluation is done only on 2D settings.
* While the final learned solutions have improved error, they are still far from machine precision, which is more important and relevant for PDE applications.
* Experimental tuning is limited (I believe only one architecture is studied without much tuning of optimization parameters).
* Presentation and exposition is very poor. For example:
  - Line 200 does not make sense, $\psi$ is both 0 and $u_0$. No conditions are given on $\phi$. This is clearly a typo but it is a very confusing one, and it greatly impacts the clarity of presentation.
  - Theorem 2 is very confusing, and there is no clear statement in the theorem. The presentation would be vastly improved by encapsulating assumptions and intermediate results into lemmas, and making the final theorem a very simple and precise statement.

**Questions:**

1. Can you summarize in simpler terms what the quantitative result of Theorem 2 actually means?
2. How well would this method generalize to higher dimensions?
3. Could this method be combined with other approaches to achieve high precision results?

---

> ### Author Response · Authors · 2025-11-12
>
> We appreciate the reviewer’s comments and the opportunity to clarify several misunderstandings.
>
> 1. On “limited experiments / only 2D”
>
> This statement is inaccurate. The paper presents results on seven distinct stiff PDEs: Allen–Cahn, Cahn–Hilliard, Kuramoto–Sivashinsky, Gray–Scott, Schrodinger, and two linear advection equations: $\textbf{covering 1D, coupled, and 2D regimes}$. Section 4 and Appendix D provide quantitative comparisons with Causal-PINN and RBA-PINN, and the 2D Allen–Cahn (Figure 2) demonstrates extension to higher dimensions. The experiments therefore go well beyond 2D “toy” problems.
>
> 2. On “errors far from machine precision”
>
> The reviewer’s expectation of “machine precision” is not aligned with the goals of PINN research.  PINNs are continuous optimization solvers; achieving machine precision is $\textbf{neither the goal nor realistic for stiff PDEs! }$ The objective of this work is to improve stability and convergence of PINNs in stiff regimes, not to match machine-precision finite-difference solvers.  As shown in Table 1, HC-PINNs achieve 1–2 orders of magnitude lower error than state-of-the-art causal and attention-based PINNs under identical budgets ($50k$ iterations). This demonstrates both stability and efficiency, which are the focus of our paper.
>
> 3. On "Experimental tuning is limited"
>
> Our objective was controlled ablation, not hyper-parameter optimization. All compared methods share identical architectures, iteration budgets, and collocation points to isolate the effect of the hard constraint. We explicitly state this fairness condition in Section 4 and Appendix D. Optimizing architectures is orthogonal to our study and left for future work.
>
> 4. On Presentation
>
> Thank you for pointing out this. The typo in Line 200 has been revised. Theorem 2 is now clearly stated, with all assumptions and intermediate derivations separated into Lemma (Loss formulations) and Lemma (Hessian expressions) before the theorem.
>
> 5.  In response to Question 1
>
> We have already made Theorem 2 clearer. See the updated PDF.
>
> 6. In response to Question 2
>
> In theory, the HC-PINN formulation is dimension-independent by construction. The transformation $ \tilde{u}(t, x) = \psi(t, x) + \phi(t, x)u_{nn}(t, x)$ and its governing equation $\phi\partial_tu_{nn} + \mathcal{P}[\psi + \phi u_{nn}] + \phi_tu_{nn} = f - \psi_t$ hold for any spatial dimension.
>
> In practice, the method already scales to 2D systems, demonstrated in Fig. 2 (Allen–Cahn 2D), where the model maintains stability and accuracy under stiff dynamics. The transformation does not introduce additional computational complexity.
>
> Importantly, we note that $\textbf{very few }$ studies in the current PINN literature have successfully trained models for high-dimensional stiff PDEs. Even baseline 3D PINNs often suffer from gradient pathologies and unstable training dynamics (see Wang et al., 2022; Kawaguchi et al., 2024). $\textbf{Our contribution is not to outperform numerical solvers in 3D,}$  $\textbf{but to clarify and mitigate the conditioning issues that make such extensions feasible in the first place.}$
>
> 7. In response to Question 3
>
> Yes. In fact, $\textbf{our approach has already been combined with several existing techniques}$ in the paper, and we include a full ablation study demonstrating this in Section 4 and Appendix D.4. HC-PINN is not an isolated architecture. It is a modular representation layer that can be integrated with various stabilization or precision-enhancing strategies.
>
> Specifically:
>
> We combine the hard-constraint transformation with spectral Fourier embeddings and adaptive residual weighting, both of which are evaluated in our ablation study.
>
> The results (see Appendix) show consistent 0.5–1 order-of-magnitude error reduction compared to either technique alone, confirming that the improvements are complementary, not redundant.
>
> Importantly, HC-PINN’s transformation does not alter the loss structure or optimizer, so it is fully compatible with more advanced frameworks. These integrations are straightforward because HC-PINN only reparametrizes the network output to enforce IC/BC exactly, but it does not constrain model design or training pipeline choices.
>
> 1. Wang, S., Sankaran, S., & Perdikaris, P. (2022). Respecting causality is all you need for training physics-informed neural networks. arXiv preprint arXiv:2203.07404.
> 2. Hu, Z., Zhang, Z., Karniadakis, G. E., & Kawaguchi, K. (2025). Score-based physics-informed neural networks for high-dimensional Fokker–Planck equations. SIAM Journal on Scientific Computing, 47(3), C680-C705.

---

### Meta-Review · Area_Chair_hx2X · 2025-12-04

**Summary:**

This paper targets the enforcement of hard-constraints with an NTK analysis for PINNS. There are definitely interesting aspects here, but at the same time, the reviewer sentiment is clearly negative (even when down-weighing one of the negative reviews).

The authors have provided detailed answers, which however only partially address some of the core concerns: the limited novelty of the direction, and the limited impact (a fundamental problem of PINNs). In addition, concerns that the paper is difficult to understand were voiced several times - there is clearly room for improvement here.

As a consequence, I recommend a reject for this paper.

**Reviewer Concerns:**

Limited novelty and limited impact are central points that were not addressed in the rebuttal.

**Reviewer Scores:**

If the reviewers had been able to participate fully in the discussion, my estimate is that it would have led to the following scores:
q5o1 2 -> 2
tpNt 6 -> 6
gfDj 2 -> 4
TEje 2 -> 2

I.e., no change for q5o1 ,tpNt , and TEje, but for gfDj I think a raise would have been possible.

---

### Decision · Program_Chairs · 2026-01-26

Reject